# Steering Large Language Models through the DMTA Cycle: Structure-Based Drug Design via Knowledge-Driven Bi-Level Thompson Sampling

Xuanning Hu [1 2]   Hao Tuo [1 2]   Jinglong Ji [3 2]   Anchen Li [4]   Qianli Xing [1 2]   Bo Yang [1 2]

## Abstract

Structure-based drug design (SBDD) can be effectively realized through an iterative refinement via the Design-Make-Test-Analyze (DMTA) cycle, which is a common workflow used by human experts. However, most conventional deep learning-based models as one-shot generators that lack feedback mechanisms, leaving the DMTA loop disconnected. In this work, we propose K-BTS, a Knowledge-Driven Bi-level Thompson Sampling framework that formalizes iterative SBDD as a Dynamic Hierarchical Multi-Armed Bandit problem. K-BTS closes the DMTA loop by decoupling decisions into two levels: an upper-level policy that prioritizes high-potential molecular lineages and a lower-level mechanism that retrieves explicit chemical rules to guide LLM generation. By integrating a dual-level Bayesian update, the framework transforms sparse docking scores into reusable experience. On the CrossDocked2020 benchmark, K-BTS achieves a state-of-the-art Top-1 average docking score. The results from diverse dimensions show that K-BTS ensures search determinism through a smooth, monotonic convergence that synchronizes structural drift with affinity improvement.

## 1. Introduction

Structure-based drug design (SBDD) aims to discover novel ligands with high binding affinity for specific protein binding sites while satisfying essential chemical and geometric constraints, and remains a central challenge in modern drug discovery. In real-world medicinal chemistry, high-affinity molecular candidates are rarely the result of a one-shot generation. By contrast, they evolve through the rigorous Design-Make-Test-Analyze (DMTA) cycle (Plowright et al., 2012; Schneider, 2018), where human experts progressively refine a lead compound based on accumulated empirical experience. Thus, capturing this evolutionary process is as critical as the final generative result.

Recent deep learning models, ranging from Variational Autoencoders (VAEs) (Chenthamarakshan et al., 2020; Ragoza et al., 2022) and autoregressive models (Luo et al., 2021; Liu et al., 2022; Peng et al., 2022) to recent diffusion-based architectures (Guan et al., 2023a;b; Huang et al., 2024; Qu et al., 2024; Gu et al., 2024), have demonstrated remarkable capabilities in generating diverse geometrically valid molecular candidates. Despite the impressive capabilities, they typically operate under a one-shot generation paradigm and lack the intrinsic mechanism to handle sequential feedback. Once a generated molecule fails to meet affinity targets, these methods cannot analyze specific structural deficiencies or perform further refinement like human experts. Consequently, they struggle to adaptively optimize candidates during the inference phase, failing to close the DMTA loop.

Large Language Models (LLMs) (Vaswani et al., 2017; Brown et al., 2020), with strong instruction-following, reasoning (Wei et al., 2022a;b), and knowledge integration abilities (Qiu et al., 2020), offer a promising paradigm to close the DMTA loop. Unlike geometric models, LLMs possess general chemical priors and the capacity for logical instruction following. Theoretically, an LLM acting as an expert can utilize natural language as an interface to process intermediate outcomes, allowing it to guide subsequent decisions based on reasoning. This ability to condition generation on past context provides a new frontier for solving long-horizon optimization problems in drug discovery.

However, autonomously deploying LLMs to execute the DMTA cycle proves non-trivial, hampered by three systemic disconnects that currently fracture the optimization loop. First, a Policy Disconnect emerges at the Analyze-to-Design phase. Existing methods (Brahmavar et al., 2024; Hu et al., 2026) lack a mechanism to distinguish low-potential

[1]College of Computer Science and Technology, Jilin University, Changchun, Jilin, China [2]Key Laboratory of Symbolic Computation and Knowledge Engineering of Ministry of Education, Jilin University, Changchun, Jilin, China [3]College of Artificial Intelligence, Jilin University, Changchun, Jilin, China [4]Department of Computer Science, Aalto University, Espoo, Finland. Correspondence to: Qianli Xing <qianlixing@jlu.edu.cn>, Bo Yang <ybo@jlu.edu.cn>.

*Proceedings of the 43rd International Conference on Machine Learning*, Seoul, South Korea. PMLR 306, 2026. Copyright 2026 by the author(s).

candidates from promising leads, treating all molecules in the pool as equally worthy of exploration or greedily explored. Second, a Translation Disconnect emerges at the Design-to-Make phase. In standard LLM generation, the specific chemical rule, e.g., "adding a hydrogen donor", is still hidden within the black-box token generation (Brahmavar et al., 2024; Hu et al., 2026). Thus, the action selection remains implicit and unobservable. Finally, the cycle is severed by a Feedback Disconnect during the Test-to-Analyze phase. The critical sparsity of scalar docking scores prevents fine-grained credit assignment, making the model unable to discern which specific modification induced an affinity shift, thereby preventing the distillation of reusable experience.

To bridge these systemic disconnects, we propose Knowledge-Driven Bi-level Thompson Sampling (K-BTS), a framework that formalizes SBDD as a Dynamic Hierarchical Multi-Armed Bandit (D-HMAB) problem to systematically harmonize the disjointed transitions of the DMTA cycle. To address the Policy Disconnect, we employ an uncertainty-aware Upper-Level Bayesian seed selection strategy. By treating each molecular lineage as a bandit arm, the system intelligently distinguishes high-potential leads from unproductive ones, optimizing resource allocation between exploration and exploitation. Simultaneously, we resolve the Translation Disconnect via Lower-Level explicit action retrieval, where generation is anchored to structured chemical rules from a dynamic knowledge base, ensuring that design intent is no longer implicit but rendered as precise, interpretable constraints. Finally, we overcome the Feedback Disconnect through a dual-level Bayesian update mechanism that decomposes scalar docking feedback into specific rules and lineage adjustments. This fine-grained credit assignment effectively transforms raw simulation data into reusable experience, thereby transforming the unguided stochastic exploration into a directed, convergent evolutionary process. The main contributions of this work are summarized as follows:

- We formally redefine LLM-based molecular optimization as a Dynamic Hierarchical Multi-Armed Bandit (D-HMAB) task, establishing a principled foundation that shifts the paradigm from static generation to rigorous sequential decision-making.

- We propose the K-BTS Framework with a bi-level Bayesian strategy that coordinates seed selection and modification retrieval, effectively transforming unfocused exploration into directed search.

- We evaluate K-BTS on the CrossDocked2020 benchmark, where it consistently outperforms eight competitive baselines, demonstrating stable optimization behavior and improved predicted binding affinity.

## 2. Problem Formulation

In LLM-driven SBDD, molecular optimization typically follows an iterative refinement paradigm that mirrors the classical DMTA cycle, starting from a set of molecules with evaluated binding affinity. In real-world medicinal chemistry, the "Make" phase traditionally refers to physical synthesis in laboratory settings. In our automated pipeline, we analogously define "Make" as the computational stage where the LLM executes specific modification intents to generate new molecular structures from previously evaluated seeds. At each iteration, the agent therefore performs a Design decision by selecting a previously evaluated molecule, followed by a Make decision that specifies how the molecule should be computationally refined, without prior knowledge of whether either choice will improve binding affinity. This naturally forms a sequential decision-making problem under uncertainty.

### 2.1. Mathematical Abstraction of the DMTA-aligned Iterative SBDD Process

Given a target protein $t \in \mathbb{T}$, SBDD aims to find ligands $m \in \mathbb{M}$ with strong binding affinity. Let $f : \mathbb{M} \times \mathbb{T} \to \mathbb{R}$ be a black-box docking function, where lower values indicate stronger binding. Given initial data $\mathcal{D}_0 = \{(m_i, f(m_i, t))\}_{i=1}^{N}$, the objective is

$$\mathcal{M}^* = \arg \min_{\mathcal{M}' \subset \mathbb{M}_{\text{rem}}, |\mathcal{M}'| = k_{cd}} \sum_{m \in \mathcal{M}'} f(m, t), \quad (1)$$

where $\mathbb{M}_{\text{rem}}$ is the unexplored chemical space and $k_{cd}$ denotes the number of final candidate molecules to be identified for downstream drug discovery. Crucially, $\mathbb{M}_{\text{rem}}$ is not directly accessible. New candidates can only be generated through sequential modification of previously evaluated molecules, enforcing a DMTA-style iterative decision process rather than one-shot selection. We therefore focus on iterative refinement under the DMTA paradigm, where new candidates are generated exclusively through sequential Design–Make decisions applied to tested molecules, and evaluated via black-box affinity feedback.

### 2.2. SBDD as a Dynamic Hierarchical Multi-Armed Bandit

We model the DMTA-style refinement process as a **Dynamic Hierarchical Multi-Armed Bandit (D-HMAB)**. At iteration $k$, the system state is the molecular pool

$$\mathcal{P}_k = \{m_1, \ldots, m_{N_k}\}, \quad (2)$$

which expands as new molecules are generated. The pool represents explored chemical space, with the goal of gradually populating it with high-affinity molecules.

Each step has a two-level decision: at the upper (Design) level, molecules are treated as arms and selecting a seed

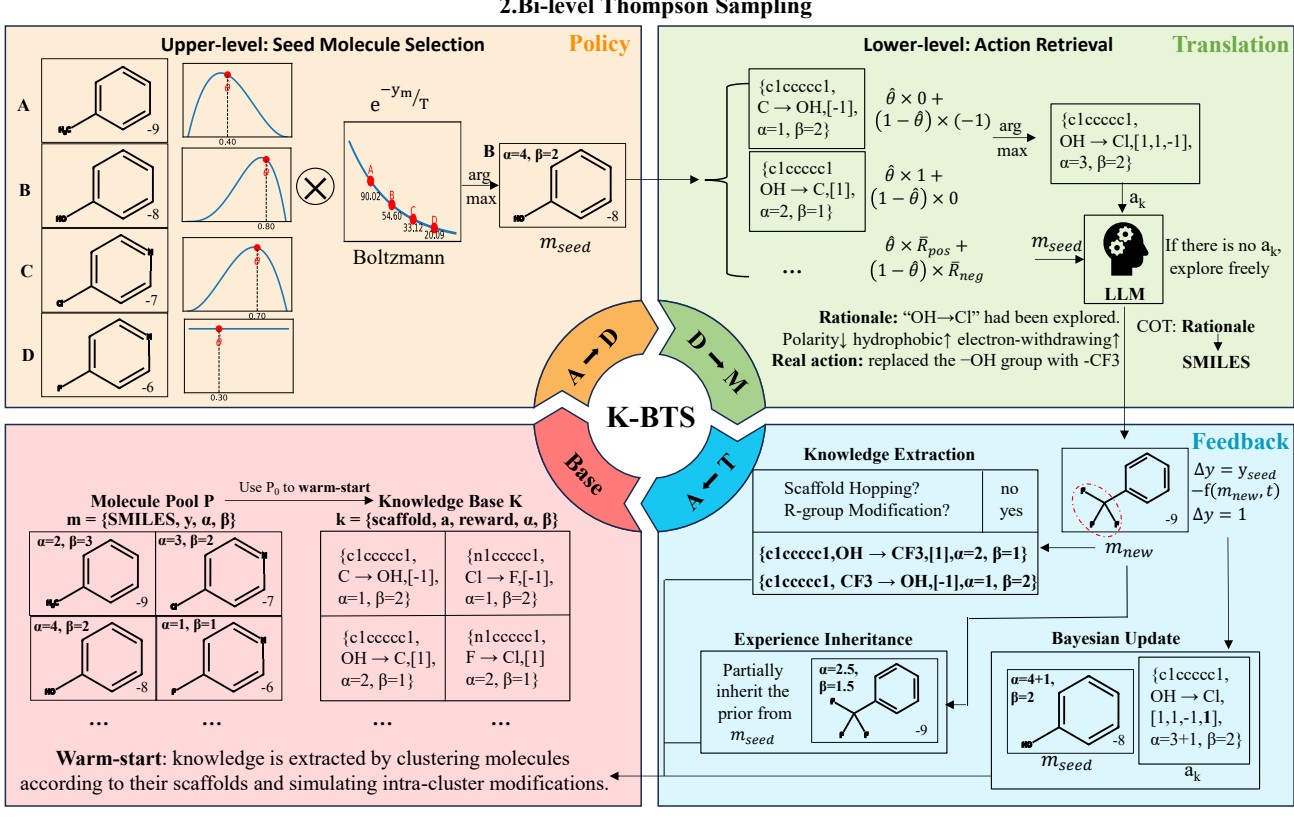

*Figure 1.* Overview of the K-BTS framework. In one iteration of K-BTS, Upper-level Thompson Sampling selects a seed molecule from the molecular pool, while Lower-level Thompson Sampling retrieves an activated modification conditioned on the seed from the knowledge base. The LLM then generates a new molecule based on the seed and modification, evaluates its performance, and updates both the molecular pool and the knowledge base.

molecule $m_{\text{seed}} \in \mathcal{P}_k$ determines the local search region; at the lower (Make) level, conditioned on the seed, the agent selects a modification intent $a_k$ from a dynamic action space $\mathcal{A}_k$. The LLM stochastically generates

$$m_{\text{new}} \sim P(m \mid m_{\text{seed}}, a_k, \mathcal{L}), \qquad (3)$$

projecting the intent onto the molecule. New modifications extracted from $m_{new}$ can expand $\mathcal{A}_k$, enabling discovery of novel strategies. After docking evaluation, the reward

$$r_k = R\big(f(m_{\text{seed}}, t) - f(m_{new}, t)\big) \qquad (4)$$

updates the agent's beliefs over seed molecules and modification intents, where $R$ is a reward function. Iteratively, this guides efficient exploration of the chemical space. Background on DMTA, multi-armed bandits, and Thompson Sampling is provided in Appendix A.

## 3. Methodology

### 3.1. Overview

Building on the D-HMAB formulation for iterative SBDD, we propose **K-BTS**, a Knowledge-driven Bi-level Thompson Sampling framework designed to close the DMTA loop and enable LLMs to refine molecules in a manner analogous to human medicinal chemists.

As illustrated in Fig. 1, K-BTS maintains two evolving components (**lower left**): the **Molecular Pool** $\mathcal{P}$, where each lineage has Beta parameters $\{\alpha_m, \beta_m\}$ to estimate refinement potential and prioritize promising seeds, and the **Knowledge Base** $\mathcal{K}$, which abstracts molecular modifications into reusable intents $a$ with parameters $\{\alpha_a, \beta_a\}$ capturing empirical effectiveness under the target binding context. Initialized from $\mathcal{D}_0$ and continuously updated, $\mathcal{K}$ closes the Feedback Disconnect. Each iteration mirrors one DMTA cycle: (**A–D, upper left**): Analyze-to-Design — select a seed $m_{\text{seed}} \in \mathcal{P}$, prioritizing high-potential lineages to address the Policy Disconnect; (**D–M, upper right**): Design-to-Make — retrieve a candidate intent $a_k \in \mathcal{K}$ as a semantic

anchor for the LLM to generate $m_{\text{new}}$, making the action explicit and observable to address the Translation Disconnect; **(T–A, lower right)**: Test-to-Analyze — docking of $m_{\text{new}}$ informs a bi-level Bayesian update, crediting both the lineage in $\mathcal{P}$ and the intent in $\mathcal{K}$, transforming sparse feedback into reusable knowledge and closing the Feedback Disconnect. This iterative process enables focused, stable, and adaptive exploration of chemical space. The complete pseudocode is in Algorithm 1 in the Appendix.

### 3.2. Knowledge Representation and Warm-start

The core of K-BTS lies in the definition of process-oriented nodes. A molecule $m \in \mathcal{P}$ is represented as $m = \{s_m, y_m, \alpha_m, \beta_m\}$, where $s_m$ denotes the molecular SMILES string, $y_m$ denotes its docking score, and $(\alpha_m, \beta_m)$ parameterize the estimated potential of $m$ as a refinement seed. A knowledge entry $\kappa \in \mathcal{K}$ encodes the empirical effect of applying a chemical action under a specific scaffold context, and is defined as:

$$\kappa = \{S_a, a, \mathcal{R}_a, \alpha_a, \beta_a\}, \quad (5)$$

where $S_a$ denotes a Murcko scaffold serving as structural context, $a$ represents a chemical action applied to this scaffold, $\mathcal{R}_a$ records normalized docking improvements from past applications of $a$, and $(\alpha_a, \beta_a)$ store accumulated success and failure statistics for that action.

To mitigate the cold-start problem, we initialize the knowledge base $\mathcal{K}$ by retrospectively extracting modification trajectories from pre-existing observations in $\mathcal{D}_0$. Specifically, molecules in $\mathcal{D}_0$ are first clustered according to their Murcko scaffolds, and the warm-start is restricted to intra-cluster permutations to ensure structural consistency. For any two molecules $m_i$ and $m_j$ sharing the same scaffold $S_a$, we define a refinement transition $\xi_{ij} : m_i \xrightarrow{a_0^{(ij)}} m_j$ whenever $|y_i - y_j| > \epsilon$, where $\epsilon$ is a margin threshold that filters out negligible docking score fluctuations and retains only meaningful performance differences.

To remain consistent with the online update mechanism, each transition $\xi_{ij}$ involving $n$ distinct substitution sites is decomposed into a set of warm-start atomic actions $\mathcal{A}_{ij}^{(0)} = \{a_{0,1}^{(ij)}, \ldots, a_{0,n}^{(ij)}\}$. Each sub-action is assigned a normalized weight $\omega_{ij} = 1/n$, reflecting its fractional contribution to the observed performance change.

Let $\Delta y_{ij} = y_i - y_j$ denote the docking score difference associated with $\xi_{ij}$. For each sub-action $a_{0,\ell}^{(ij)}$, we initialize its Bayesian parameters according to the sign of $\Delta y_{ij}$:

$$(\alpha_{a_{0,\ell}^{(ij)}}, \beta_{a_{0,\ell}^{(ij)}}) = \begin{cases} (1 + \omega_{ij}, 1), & \text{if } \Delta y_{ij} > 0, \\ (1, 1 + \omega_{ij}), & \text{if } \Delta y_{ij} < 0, \end{cases} \quad (6)$$

and append the normalized reward $\Delta y_{ij}/n$ to $\mathcal{R}_{a_{0,\ell}^{(ij)}}$.

Finally, each sub-action is combined with its corresponding scaffold $S_a$ to form a knowledge entry $\kappa = \{S_a, a_{0,\ell}^{(ij)}, \mathcal{R}_{a_{0,\ell}^{(ij)}}, \alpha_{a_{0,\ell}^{(ij)}}, \beta_{a_{0,\ell}^{(ij)}}\}$, and all entries extracted from $\mathcal{D}_0$ are aggregated into the knowledge base $\mathcal{K}$.

### 3.3. Bi-level Thompson Sampling Strategy

#### 3.3.1. UPPER-LEVEL: SEED MOLECULE SELECTION

The agent begins each iteration by selecting a seed molecule $m_{seed}$ from the current pool $\mathcal{P}_k$. This selection must balance the current binding affinity with the refinement potential, which is formalized via a selection score $S(m)$:

$$S(m) = \hat{\theta}_m \cdot \exp(-y_m/T), \quad \hat{\theta}_m \sim \text{Beta}(\alpha_m, \beta_m), \quad (7)$$

where $y_m$ and $T$ denote the docking score and temperature, respectively. Smaller values of $T$ place stronger emphasis on molecules with lower docking scores during seed selection. The optimal seed molecule is then determined by maximizing this score across the pool:

$$m_{\text{seed}} = \underset{m \in \mathcal{P}_k}{\arg\max} \, S(m). \quad (8)$$

The exponential term implements a Boltzmann distribution to favor molecules with low docking scores, while the sampled success probability $\hat{\theta}_m$ introduces a Bayesian belief component. This mechanism reinforces the selection of productive lineages with high $\alpha_m$ while encouraging the exploration of under-refined candidates. By integrating these components, K-BTS ensures that computational resources are concentrated on molecules with high refinement potential, effectively avoiding local energy minima.

#### 3.3.2. LOWER-LEVEL: ACTION RETRIEVAL

For the selected $m_{\text{seed}}$, the agent retrieves an action $a_k$ by maximizing the expected utility $U(a)$, which weighs potential affinity gains against the risk of failure:

$$U(a) = \hat{\theta}_a \bar{R}_{pos} + (1 - \hat{\theta}_a)\bar{R}_{neg}, \quad \hat{\theta}_a \sim \text{Beta}(\alpha_a, \beta_a), \quad (9)$$

where $\bar{R}_{pos}$ and $\bar{R}_{neg}$ denote historical average positive and negative rewards. To ensure structural relevance, we first find the most similar known scaffold $s^* = \arg\max_s \text{Sim}(\text{MS}(m_{seed}), s)$. A gated selection rule is then applied:

$$a_k = \begin{cases} \underset{a \in \mathcal{A}_{s^*}}{\arg\max} U(a), & \text{if } \text{Sim}(\text{MS}(m_{\text{seed}}), s^*) \geq \tau \\ \text{None}, & \text{otherwise}, \end{cases}, \quad (10)$$

where $\tau$ is a confidence threshold. If the maximum scaffold similarity falls below $\tau$, the agent bypasses the knowledge base $\mathcal{K}$ and lets the LLM explore freely.

### 3.3.3. GENERATION VIA KNOWLEDGE-DRIVEN PRUNING

Following the retrieval of $m_{\text{seed}}$ and $a_k$, the LLM acts as the refinement engine to generate the refined candidate $m_{\text{new}}$. In this framework, $a_k$ functions as a semantic anchor that constrains the LLM's vast generative space. To ensure transparency, we employ a single-step CoT strategy:

$$Rationale, m_{\text{new}} = \text{LLM}(\text{Prompt}(m_{\text{seed}}, a_k, \mathcal{H}_m)), \tag{11}$$

where $\mathcal{H}_m$ is a subset of molecules in $\mathcal{P}_k$ possessing the highest structural similarity to $m_{\text{seed}}$. The prompt instructs the LLM to first output a Rationale followed by the modified SMILES. The prompt instructs the LLM to first output a Rationale followed by the modified SMILES. Specifically, the prompt contains four components: (i) the current lead molecule and docking context, (ii) a set of forbidden previously evaluated derivatives, (iii) strategic inspiration derived from the retrieved knowledge action $a_k$, and (iv) explicit generation constraints and output formatting requirements. The complete template is provided in Appendix C.2.

By explicitly prohibiting regeneration of molecules in $\mathcal{H}_m$, the prompt forces the LLM to avoid simple structural replication. This encourages the LLM to infer the pharmacophoric intent of $a_k$ and perform reasoning within a **local action space**. For example, if $a_k$ aims to increase hydrophilicity but existing derivatives in $\mathcal{H}_m$ have exhausted certain polar substitutions, the LLM identifies alternative implementations, such as different bioisosteric replacements or novel polar group placements, compatible with $m_{\text{seed}}$ yet distinct from $\mathcal{H}_m$.

### 3.4. Knowledge Extraction and Bayesian Update

#### 3.4.1. POOL INTEGRATION AND EXPERIENCE INHERITANCE

The new candidate $m_{\text{new}}$ is integrated into $\mathcal{P}_{k+1}$ by inheriting a "softened" prior from its parent $m_{\text{seed}}$. To balance the lineage bias with the need for exploration, we update its parameters using an inheritance constant $\gamma \in (0, 1)$:

$$\alpha_{m_{\text{new}}} = 1 + \gamma(\alpha_{m_{\text{seed}}} - 1), \quad \beta_{m_{\text{new}}} = 1 + \gamma(\beta_{m_{\text{seed}}} - 1). \tag{12}$$

This mechanism allows $m_{\text{new}}$ to inherit empirical successes from its lineage while its higher posterior variance promotes immediate validation under Thompson Sampling.

#### 3.4.2. BIDIRECTIONAL KNOWLEDGE EXTRACTION AND UPDATE

For each modification applied by the LLM from the seed molecule $m_{\text{seed}}$ to the new molecule $m_{\text{new}}$, we first compute the observed docking score change $\Delta y = y_{\text{seed}} - y_{\text{new}}$. The modification is represented bidirectionally: a forward action $a_f : m_{\text{seed}} \to m_{\text{new}}$ with reward $r_f = \Delta y$, and a backward action $a_b : m_{\text{new}} \to m_{\text{seed}}$ with reward $r_b = -\Delta y$, capturing both improvement and deterioration effects.

If the scaffold changes, the entire modification is treated as a single action with weight $\omega = 1$. If the scaffold remains the same, the transition is decomposed into $n$ atomic sub-actions corresponding to the $n$ modified sites on the scaffold, each assigned a weight $\omega = 1/n$ and reward $r = \Delta y/n$ (or $r = -\Delta y/n$ for the backward action). All extracted actions from the current iteration are collected into a set $\mathcal{A}_{\text{new}}$. For any action in $\mathcal{A}_{\text{new}}$ not already present in the knowledge base $\mathcal{K}$, we initialize its Beta distribution parameters as $(\alpha_a, \beta_a) = (1, 1)$. Next, each action is combined with its corresponding scaffold to form a knowledge entry: the forward action is associated with the scaffold of $m_{\text{seed}}$, and the backward action with the scaffold of $m_{\text{new}}$. These entries are then added to the knowledge base $\mathcal{K}$.

Finally, the parameters of all actions in $\mathcal{A}_{\text{new}}$, both new and existing, are updated according to

$$(\alpha_a, \beta_a) \leftarrow \begin{cases} (\alpha_a + \omega, \beta_a), & r > 0, \\ (\alpha_a, \beta_a + \omega), & \text{otherwise}, \end{cases} \tag{13}$$

ensuring that each action receives a credit update corresponding to its actual observed effect, whether it is newly extracted or previously existing, even if it was not selected as a guidance action in the main iteration.

#### 3.4.3. BI-LEVEL BAYESIAN PARAMETER UPDATE

The final stage refines the parameters of nodes in the current iteration to reflect the optimization outcome. To explicitly account for the **optimization magnitude**, so that molecules yielding substantial affinity improvements receive higher selection priority than those producing marginal gains, the **upper-level update** adjusts the seed molecule's parameters via a magnitude-sensitive weight $\omega_m = \lambda \cdot |\Delta y|$:

$$(\alpha, \beta)_{m_{\text{seed}}} \leftarrow \begin{cases} (\alpha + \omega_m, \beta), & \text{if } \Delta y > 0, \\ (\alpha, \beta + \omega_m), & \text{if } \Delta y \leq 0 \end{cases}, \tag{14}$$

where $\lambda$ is a scaling factor.

Conversely, the **lower-level update** treats the action $a_k$ as a Bernoulli trial to evaluate its general applicability:

$$(\alpha, \beta)_{a_k} \leftarrow \begin{cases} (\alpha + 1, \beta), & \text{if } \Delta y > 0 \\ (\alpha, \beta + 1), & \text{if } \Delta y \leq 0 \end{cases}. \tag{15}$$

### 3.5. Theoretical Intuition

We provide a high-level intuition for the asymptotic improvement of K-BTS. Detailed assumptions and supporting arguments are deferred to Appendix B.

**Proposition 3.1** (Asymptotic Improvement Tendency of K-BTS). *Under mild regularity assumptions on the docking landscape and the generative policy (see Appendix B.1), K-BTS exhibits an asymptotic improvement tendency. In particular, for any $\epsilon > 0$, there exists $K_\epsilon$ such that for all $k \geq K_\epsilon$*

$$\mathbb{P}\Big( f(m_{\text{best}}^{(k)}) - f(m^*) < \epsilon \Big) \qquad (16)$$

*tends to increase as exploration proceeds, and approaches 1 asymptotically.*

This behavior can be intuitively attributed to the combination of (i) Thompson Sampling at the lower level, which increasingly favors beneficial modifications, and (ii) uncertainty-aware seed selection at the upper level, which balances exploitation of promising molecular lineages with sustained exploration (see Appendix B.2).

# 4. Experiments

## 4.1. Experimental Setup

### 4.1.1. DATASET AND SETTINGS

We evaluate **K-BTS** on the CrossDocked2020 dataset (Francoeur et al., 2020), following the experimental settings established by ELILLM (Hu et al., 2026). The dataset comprises 65,000 drug–target pairs for training and 100 targets for testing. Consistent with their setup, we construct the initial observed dataset $\mathcal{D}_0$ under two settings to assess refinement performance from different chemical starting points: (1) **K-BTS-rand**, which initializes from 100 ligands randomly sampled from the training set; and (2) **K-BTS-diff**, which utilizes molecules and docking scores generated by the SOTA diffusion model, ALIDiff (Gu et al., 2024). Our framework accesses **DeepSeek-V3.2** (Liu et al., 2025) via API as the basic LLM, leveraging its reasoning capabilities without the need for parameter updates. Our code is available at https://github.com/hxnhxn/K-BTS

### 4.1.2. BASELINES

We compare K-BTS against three categories of SBDD methods: (1) **Structure-based Generative Models**, comprising the voxel-based VAE **liGAN** (Ragoza et al., 2022) and the 3D atomic-level autoregressive models **AR** (Luo et al., 2021) and **Pocket2Mol** (Peng et al., 2022); (2) **Diffusion Models**, including **TargetDiff** (Guan et al., 2023a) and the SOTA **ALIDiff** (Gu et al., 2024); (3) **LLM-based Models**, including **TamGen** (Wu et al., 2024) (a GPT-based chemical language model trained for de-novo conditional generation), as well as the iterative refinement models **LMLF** (Brahmavar et al., 2024) and **ELILLM** (Hu et al., 2026).

To ensure a rigorous head-to-head comparison, **K-BTS**, **LMLF**, and **ELILLM** are evaluated in both **-rand** and **-diff**

variants. Following its original implementation, **ELILLM** uses a locally deployed **LLaMA-3.1-8B** for latent space exploration, where optimization is performed directly on the model's latent space. In contrast, **LMLF** is implemented via the **same DeepSeek-V3.2 API** as K-BTS.

### 4.1.3. EVALUATION METRICS

Consistent with ELILLM (Hu et al., 2026), we generate 100 candidates per test target and prioritize the **Vina docking score** (Trott & Olson, 2010) (calculated via smina (Koes et al., 2013)) as the primary optimization objective. We also report **QED** (Bickerton et al., 2012), **SA** (Ertl & Schuffenhauer, 2009), and **Internal Diversity** as reference metrics. Following ELILLM, all molecules are standardized with RDKit 3D reconstruction and force-field optimization to ensure consistent Vina scores across identical SMILES. Further experimental details are provided in Appendix D.

## 4.2. Binding Affinity Evaluation

Table 1 summarizes the binding affinity and molecular quality metrics on the CrossDocked2020 test set. Our proposed **K-BTS** framework consistently achieves state-of-the-art (SOTA) performance across all docking-related metrics.

**Quantitative Lead via D-HMAB Policy.** As demonstrated in the results, **K-BTS-diff** delivers the most competitive performance, reaching a Top-1 average Vina score of **-10.23**. This represents a substantial improvement over the strongest iterative baseline, ELILLM-diff (-9.80), and significantly outperforms LMLF-diff (-9.29). Crucially, the performance margin underscores the efficacy of the **D-HMAB strategy**. By decoupling the exploration into a hierarchical selection of lineages and knowledge-retrieved actions, D-HMAB effectively orchestrates the LLM to navigate complex binding landscapes more efficiently than standard iterative refinement. Notably, the Top-20 average docking score of K-BTS-diff (**-9.44**) surpasses the Top-1 average scores of the SOTA diffusion model ALIDiff (-9.37). This proves that our policy does not merely optimize for individual outliers but systematically shifts the entire candidate pool toward higher binding potentials. These gains are supported by one-sided paired Wilcoxon tests with Holm–Bonferroni correction (all corrected p-values $< 2.01 \times 10^{-7}$; Additional results are provided in Appendix E.1 and Appendix E.2.

**Adaptability from Pocket-Agnostic Starting Points.** A key highlight of our framework is its robustness under suboptimal starting conditions. **K-BTS-rand**, which initializes from a set of 100 *pocket-agnostic* ligands sampled from the training set, achieves a Top-1 average score of -9.60. Remarkably, this performance exceeds that of specialized, full-parameter trained generative models such as TargetDiff (-9.38) and ALIDiff (-9.37). This outcome validates that K-BTS can effectively "re-purpose" general chemical scaffolds

*Table 1.* Performance Comparisons on Vina Docking Scores. We report both average (Avg.) and median (Med.) values across targets. Med. denotes the mean of per-target medians for most metrics, and the median of target-level scores for diversity. Best and second-best results are shown in **bold** and underlined, respectively.

| Methods | Top1 Dock ↓ | | Top5 Dock↓ | | Top10 Dock↓ | | Top20 Dock↓ | | QED | | SA | | Diversity | |
|---|---|---|---|---|---|---|---|---|---|---|---|---|---|---|
| | Avg. | Med. | Avg. | Med. | Avg. | Med. | Avg. | Med. | Avg. | Med. | Avg. | Med. | Avg. | Med. |
| Random Init | -8.62 | -8.62 | -8.19 | -8.14 | -7.90 | -7.81 | -6.38 | -7.42 | 0.53 | 0.53 | 0.76 | 0.77 | 0.78 | 0.77 |
| AR (2021) | -8.41 | -8.41 | -8.08 | -8.03 | -7.88 | -7.82 | -7.64 | -7.56 | 0.51 | 0.50 | 0.63 | 0.63 | 0.70 | 0.70 |
| Pocket2Mol (2022) | -9.15 | -9.15 | -8.85 | -8.82 | -8.64 | -8.60 | -8.38 | -8.32 | 0.56 | 0.57 | 0.74 | 0.75 | 0.69 | 0.71 |
| liGAN (2022) | -8.13 | -8.13 | -7.78 | -7.73 | -7.58 | -7.53 | -7.34 | -7.26 | 0.39 | 0.39 | 0.59 | 0.57 | 0.66 | 0.67 |
| TargetDiff (2023) | -9.38 | -9.38 | -8.86 | -8.78 | -8.53 | -8.43 | -8.13 | -8.00 | 0.48 | 0.48 | 0.58 | 0.58 | 0.72 | 0.71 |
| ALIDIFF (2024) | -9.37 | -9.37 | -8.85 | -8.80 | -8.52 | -8.40 | -8.11 | -8.01 | 0.50 | 0.50 | 0.57 | 0.56 | 0.73 | 0.71 |
| TamGen (2024) | -8.53 | -8.53 | -8.11 | -8.05 | -7.72 | -7.65 | -7.70 | -7.62 | 0.56 | 0.56 | 0.77 | 0.78 | 0.75 | 0.74 |
| LMLF-rand (2024) | -8.66 | -8.66 | -8.19 | -8.12 | -7.91 | -7.83 | -7.56 | -7.45 | 0.57 | 0.58 | 0.77 | 0.78 | 0.80 | 0.79 |
| ELILLM-rand (2026) | -9.33 | -9.33 | -8.98 | -8.93 | -8.74 | -8.68 | -8.44 | -8.36 | 0.46 | 0.45 | 0.63 | 0.63 | 0.69 | 0.70 |
| **K-BTS-rand (ours)** | -9.60 | -9.60 | -9.34 | -9.29 | -9.18 | -9.13 | -9.00 | -8.95 | 0.53 | 0.53 | 0.75 | 0.75 | 0.55 | 0.55 |
| LMLF-diff (2024) | -9.29 | -9.29 | -8.76 | -8.67 | -8.44 | -8.34 | -8.08 | -7.97 | 0.54 | 0.54 | 0.63 | 0.63 | 0.76 | 0.74 |
| ELILLM-diff (2026) | -9.80 | -9.80 | -9.37 | -9.31 | -9.09 | -9.03 | -8.74 | -8.65 | 0.49 | 0.49 | 0.57 | 0.56 | 0.67 | 0.68 |
| **K-BTS-diff (ours)** | **-10.23** | **-10.23** | **-9.89** | **-9.83** | **-9.68** | **-9.62** | **-9.44** | **-9.37** | 0.53 | 0.53 | 0.58 | 0.58 | 0.61 | 0.63 |

into high-affinity binders for specific targets, successfully bridging the gap between target-unrelated initialization and high-precision molecular docking.

**Molecular Plausibility and Quality.** Regarding auxiliary metrics, K-BTS maintains a balanced profile between binding potency and structural integrity. A key observation is that in the **-rand** setting, K-BTS preserves a high **average SA score of 0.75**, which is notably higher than that of ELILLM-rand (0.63). This suggests that when starting from pocket-agnostic scaffolds, our retrieved action heuristics provide crucial "synthetic guidance," steering the LLM to execute structural modifications that are not only effective for binding but also maintain high synthetic accessibility. In the **-diff** setting, while the SA score gain is more moderate due to the already optimized nature of the diffusion-generated starting points, K-BTS still maintains a competitive *QED* score (0.53) within a reasonable range for drug-like molecules. These results confirm that the enhanced affinity achieved by K-BTS does not come at the cost of overall molecular quality or chemical feasibility.

### 4.3. Iterative Evolution Analysis

To further investigate the dynamic optimization process, we compare the iterative trajectories of K-BTS-rand against iterative baselines. Unlike conventional cumulative metrics that report the best-found scores up to a certain step, we evaluate the instantaneous performance by calculating metrics independently for each batch of 10 molecules generated at each interval. This setting provides a clearer observation of the models' immediate optimization behaviors and evolutionary trends. We performed a similar analysis for the -diff setting, which exhibits trajectories highly consistent with

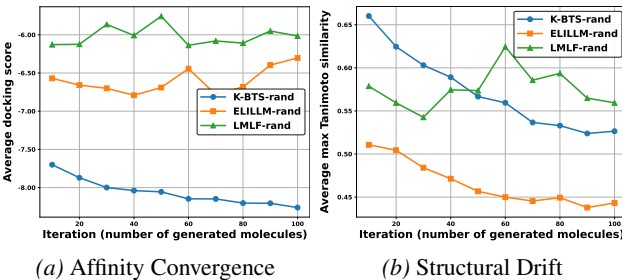

*(a)* Affinity Convergence      *(b)* Structural Drift

*Figure 2.* Iterative evolution analysis across 100 targets. (a) Average docking score and (b) Average maximum Tanimoto similarity to $\mathcal{D}_0$, both calculated independently for each batch of 10 molecules to capture instantaneous optimization trends.

the -rand setting, as shown in Appendix E.3.

**Affinity Convergence and Search Determinism.** Fig. 2a shows the evolution of average docking affinity across 100 targets. K-BTS-rand exhibits a smooth, monotonic, and rapid decline in docking scores. In contrast, baselines such as ELILLM-rand show fluctuating, largely stagnant trajectories, reflecting less structured and less deterministic optimization. This result aligns with our theoretical intuition (Section B), showing that K-BTS consistently drives the population toward superior binding manifolds at each iteration rather than relying on lucky local samples.

**Exploration Capability and Structural Drift.** Fig. 2b shows the structural similarity relative to the initial set $\mathcal{D}_0$, calculated for each batch. ELILLM-rand maintains the lowest similarity throughout, indicating aggressive exploration of chemical space. Correlated with its affinity curve, this suggests stochastic jumps lacking directional guidance. In contrast, K-BTS-rand shows a steady, purposeful decline in similarity, reflecting a knowledge-guided evolutionary pro-

*Table 2.* The performance comparisons of K-BTS-rand variants.

| Methods | Top1 Dock ↓ | | Top20 Dock↓ | | *Diversity* | |
|---|---|---|---|---|---|---|
| | Avg. | Med. | Avg. | Med. | Avg. | Med. |
| K-BTS-rand | **-9.83** | **-9.83** | **-9.14** | **-9.09** | 0.50 | 0.48 |
| w/o upper | -8.95 | -8.95 | -7.79 | -7.68 | 0.76 | 0.75 |
| w/o lower | -9.61 | -9.61 | -8.94 | -8.86 | 0.57 | 0.56 |
| w/o knowledge | -9.54 | -9.54 | -8.83 | -8.77 | 0.56 | 0.54 |
| w/o warm-start | -9.74 | -9.74 | -9.11 | -9.03 | 0.50 | 0.50 |
| Random Init | -8.58 | -8.58 | -7.71 | -7.64 | 0.78 | 0.77 |

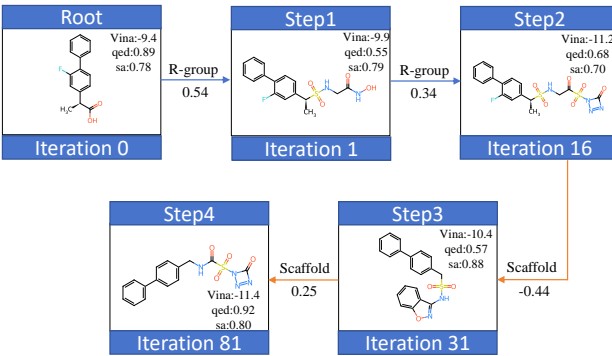

*Figure 3.* The case study of target 4PXZ under -rand setting.

cess where structural drift aligns with affinity improvement. This stability demonstrates that K-BTS navigates the chemical manifold toward superior regions in a predictable and robust manner, bridging the gap between pocket-agnostic initialization and high-precision docking.

### 4.4. Ablation Study

To decompose the contribution of each component within the K-BTS framework, we conduct an ablation study on the first 10 targets of the test set. All iterative variants are initialized from the same Random Init set ($\mathcal{D}_0$). As summarized in Table 2, the removal of any modular component leads to performance degradation, confirming the synergistic effect of our integrated design. Similar conclusions are observed under the **-diff** setting, with the corresponding ablation results reported in Appendix E.4.

**The Dominance of Upper-Level Seed Selection.** The critical performance regression is observed in the w/o upper variant, where the Top-1 average score retreats to -8.95 kcal/mol, closely approaching the Random Init baseline. This result isolates the unique contribution of the upper-level policy: strategic resource allocation. Without the ability to prioritize high-potential lineages, the search process degenerates into an unguided walk. Consequently, the downstream effect is that the retrieved lower-level knowledge is squandered on unproductive seeds. This proves that the efficacy of lower-level actions is strictly contingent upon the strategic focus established by the upper-level selection.

Notably, the full K-BTS model exhibits lower diversity (0.50) compared to Random Init (0.78) or w/o upper (0.76).

*Table 3.* Wall-clock time breakdown of K-BTS under different experimental settings. All times are reported in hours.

| Setting | TS Time | LLM Time | Evaluation Time |
|---|---|---|---|
| K-BTS-rand | 0.46 | 65.55 | 79.38 |
| K-BTS-diff | 0.69 | 60.60 | 29.68 |

This reduction is not a loss of exploratory power but a direct consequence of intentional convergence. By identifying and intensively exploiting elite chemical lineages, K-BTS effectively shifts the molecular population from a scattered random distribution toward a focused set of high-affinity candidates, demonstrating a highly efficient search-to-optimization transition.

**The Importance of Knowledge Base.** The comparison between w/o lower and w/o knowledge isolates and validates the intrinsic quality of our constructed Knowledge Base. Even when the strategic action selection is removed, meaning structural actions are applied stochastically from the retrieved pool, the w/o lower variant consistently outperforms the direct free exploration of the w/o knowledge baseline. This result is pivotal: it demonstrates that the retrieved knowledge manifold is inherently "dense" with productive chemical rules. Consequently, the mere availability of these curated actions provides superior directional guidance compared to unconstrained LLM generation, proving that the performance gain stems fundamentally from the high relevance of the retrieved knowledge itself, rather than solely from the action selection algorithm. More detailed analyses on the effects of knowledge retrieval and warm-start initialization are provided in Appendix E.4.

### 4.5. Case Study

We conduct a representative case study on target 4PXZ to visualize the navigation process of K-BTS in a complex chemical space. As illustrated in Fig. 3, K-BTS demonstrates a balanced exploitation–exploration behavior. It first performs R-group substitution on the root molecule, followed by constrained R-group reconfiguration to improve binding affinity. When the local chemical neighborhood becomes saturated, K-BTS applies scaffold hopping, causing a temporary performance drop and deprioritization of that branch. Benefiting from its exploration capability, this branch is later revisited, and retrieval-inspired recombination yields the final best-performing molecule. A detailed step-by-step analysis of the molecular evolutions is provided in Appendix E.5. In addition, Appendix E.6 presents a visualization of the docking pose of the Step 4 molecule.

### 4.6. Computational Time Analysis

As shown in Table 3 and Fig. 4, we report a detailed breakdown of the computational cost of K-BTS under the **-rand** and **-diff** settings. Specifically, the **-rand** experiments were

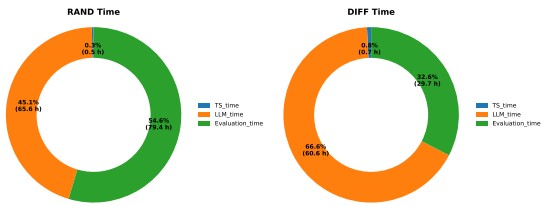

*Figure 4.* Time composition of K-BTS under different settings. Left: **-rand**. Right: **-diff**.

performed on a server equipped with an **Intel(R) Xeon(R) Silver 4310 CPU @ 2.10GHz**, while the **-diff** experiments were conducted on a server equipped with an **Intel(R) Xeon(R) Gold 6430** processor.

It can be observed that the **-diff** setting exhibits a shorter overall runtime, which is primarily attributable to the more powerful CPU architecture. Since docking evaluation is CPU-intensive, improved hardware efficiency leads to a reduced absolute evaluation time and, consequently, a lower proportion of evaluation cost in the overall runtime.

Notably, the overhead introduced by Thompson Sampling is negligible in both settings, accounting for less than **1%** of the total runtime. This indicates that the bandit-based decision module itself does not constitute a substantial computational burden, while LLM-based generation and docking-based evaluation dominate the overall runtime. In particular, docking evaluation remains one of the main efficiency bottlenecks due to its CPU-intensive nature. A promising direction for further acceleration is therefore to incorporate deep learning-based molecular representation (Li et al., 2024; 2025) and property prediction models (Li et al., 2021; Lim et al., 2025) as low-fidelity surrogate evaluators, which can pre-screen generated candidates before expensive docking calculations.

## 5. Related Work

The DMTA cycle is a long-established paradigm in medicinal chemistry, where drug discovery is treated as an iterative optimization process guided by successive structural refinement and feedback (Plowright et al., 2012). In contrast, most existing deep generative models for structure-based drug design, including VAEs, autoregressive models, and diffusion-based architectures, operate in a single-step generation regime (Ragoza et al., 2022; Luo et al., 2021; Peng et al., 2022; Guan et al., 2023a; Huang et al., 2024; Gu et al., 2024; Qu et al., 2024). While effective at producing diverse and geometrically valid molecules, these methods generally lack mechanisms for iterative refinement and adaptive feedback integration. More recently, LLMs have been explored as a further step toward iterative molecular design, leveraging their instruction-following, reasoning, and knowledge integration capabilities to incorporate external evaluation

signals across iterations (Fang et al., 2023; Brahmavar et al., 2024; Gao et al., 2025; Hu et al., 2026).

Despite recent progress, autonomously executing the DMTA cycle remains challenging due to three recurring disconnects. At a closer look, these issues reflect more general difficulties in sequential decision-making and long-horizon optimization. The first concerns trajectory selection, where systems struggle to prioritize promising molecular lineages under limited evaluation budgets. The second relates to action abstraction, as structural modifications are often generated implicitly without explicit or interpretable decision representations. The third arises from sparse feedback, where scalar evaluation signals provide insufficient guidance for attributing outcomes to individual decisions. Together, these challenges limit effective experience accumulation and hinder sustained optimization.

## 6. Conclusion

In this paper, we propose K-BTS, a framework that formalizes SBDD as a D-HMAB problem, effectively bridging the three critical disconnects of the DMTA and guiding LLMs to close the loop. Our approach transforms the Design and Make phases by coordinating uncertainty-aware seed selection with explicit modification retrieval, ensuring that LLM-driven generation is both strategic and interpretable. Furthermore, by implementing a bi-level Bayesian update mechanism during the Test and Analyze phases, K-BTS effectively distills reusable chemical knowledge from sparse docking feedback, enabling a directed evolutionary search rather than unguided stochastic exploration. Comprehensive experiments on the CrossDocked2020 benchmark validate that K-BTS achieves a systemic elevation of molecular quality rather than isolated successes. The framework reaches a state-of-the-art average Top-1 docking score over existing iterative and diffusion-based baseline models. Notably, the Top-20 average scores of K-BTS consistently outperform the competitive baselines, proving its ability to shift the entire distribution of the candidate pool toward higher binding potentials. This robust performance is underpinned by the synergy of closing the DMTA loop: strategic seed selection ensures search determinism, while knowledge-guided action refinement maintains high synthetic accessibility and structural plausibility.

## Acknowledgements

This work is supported by the National Natural Science Foundation of China under Grant Nos. U22A2098 and 62406127, the KeyScience and Technology Development Plan of Jilin Province under Grant No.20240302078GX, Major Science and Technology Project of Changchun City (2024WX05).

## Impact Statement

This paper presents a methodological contribution to advancing Machine Learning techniques for structure-based drug design, with the goal of improving the efficiency and stability of iterative molecular optimization. By formalizing the Design–Make–Test–Analyze process within a principled learning framework, our work aims to support computational drug discovery workflows and reduce the cost of early-stage molecular exploration.

Potential positive societal impacts include accelerating the identification of high-affinity drug candidates and lowering barriers to applying advanced optimization techniques in medicinal chemistry. At the same time, the proposed framework operates entirely on abstract molecular representations and docking-based evaluations, without direct involvement in clinical decision-making, patient data, or biological experimentation. As such, it does not introduce new risks beyond those commonly associated with computational drug design research.

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

# A. Preliminaries and Background

In this appendix, we provide a detailed overview of the Design-Make-Test-Analyze (DMTA) process in structure-based drug design, as well as foundational concepts from multi-armed bandits (MABs) and Thompson Sampling (TS). This background clarifies how our K-BTS framework generalizes classical methods and incorporates domain knowledge for efficient molecular optimization.

## A.1. The DMTA Cycle in Drug Discovery

The **Design-Make-Test-Analyze (DMTA)** cycle (Plowright et al., 2012; Schneider, 2018) is a core iterative workflow in medicinal chemistry:

1. **Design:** Generate candidate molecules based on prior knowledge of the target protein and the chemical space explored so far.

2. **Make:** Synthesize or simulate the proposed candidates, producing molecules that can be physically or computationally evaluated.

3. **Test:** Measure relevant properties, such as docking scores, binding affinities, or other biochemical assays.

4. **Analyze:** Interpret the results to inform the next round of design, identifying promising structural motifs and learning from unsuccessful modifications.

High-affinity molecular candidates rarely emerge from a single-shot generation. Instead, they evolve through multiple iterations, where human experts iteratively refine scaffolds and substituents, accumulate empirical insights, and explore local chemical neighborhoods. Capturing this iterative, feedback-driven process is critical for practical success. K-BTS explicitly formalizes this iterative refinement within a principled learning framework, translating DMTA decisions into sequential actions in a dynamic optimization problem.

## A.2. Multi-Armed Bandits (MABs)

A **multi-armed bandit (MAB)** problem (Robbins, 1952) models sequential decision-making under uncertainty. At each time step $t$, an agent selects an *arm* $a_t$ from a set $\mathcal{A}$ and observes a reward $r_t(a_t)$. The goal is to maximize the cumulative reward over $T$ steps, or equivalently, minimize the cumulative regret:

$$R_T = \sum_{t=1}^{T} \left( r^* - r_t(a_t) \right), \quad r^* = \max_{a \in \mathcal{A}} \mathbb{E}[r(a)]. \tag{17}$$

A core challenge in MABs is balancing **exploration** (trying uncertain arms to discover high rewards) with **exploitation** (choosing arms known to yield high rewards). Classical strategies include $\epsilon$-greedy policies, Upper Confidence Bound (UCB) methods, and Bayesian approaches such as Thompson Sampling.

## A.3. Thompson Sampling (TS)

Thompson Sampling was originally proposed earlier by **Thompson (1933)** (Thompson, 1933) in a general probabilistic context, before the formal multi-armed bandit problem was introduced by **Robbins (1952)** (Robbins, 1952). It was later recognized as a natural Bayesian strategy to address MAB problems.

**Thompson Sampling (TS)** (Thompson, 1933; Chapelle & Li, 2011) is a Bayesian MAB strategy that naturally balances exploration and exploitation:

- Each arm $a$ maintains a posterior distribution over its expected reward, $P(\theta_a \mid \mathcal{H}_t)$, where $\mathcal{H}_t$ is the history of actions and observed rewards.

- At each time step, a sample $\hat{\theta}_a$ is drawn from each arm's posterior.

- The arm with the highest sampled reward is selected.

- Observed rewards are used to update the posterior distributions.

TS has several desirable properties: it automatically favors arms with high uncertainty, it achieves near-optimal regret in many settings, and it generalizes to structured or hierarchical arm sets, making it suitable for complex decision problems.

### A.4. Relation to K-BTS

While classical MAB and TS operate on *fixed arm sets*, molecular optimization introduces several additional challenges:

- **Dynamic Upper-Level Arms:** Seed molecules in the pool are discovered and evaluated progressively; new promising molecules are continuously added, while low-potential ones can be deprioritized.

- **Dynamic Lower-Level Arms:** Modification actions (e.g., substituent changes, scaffold modifications) are discovered incrementally from previously generated molecules, forming a growing action space.

- **Integration with DMTA:** Each Design–Make–Test–Analyze iteration corresponds to an upper-level arm selection (Design) followed by a lower-level arm selection (Make), with rewards (Test) updating both levels and insights (Analyze) shaping subsequent exploration.

K-BTS extends classical TS to handle these *dynamic hierarchical arms*:

1. The **upper-level TS** selects seed molecules by considering both expected docking scores and uncertainty-aware potential for further refinement, addressing the Policy Disconnect.

2. The **lower-level TS** selects specific modification actions from a dynamic knowledge base, making design intent explicit and interpretable, addressing the Translation Disconnect.

3. A **bi-level Bayesian update** decomposes scalar docking feedback into actionable lineage and rule-level information, addressing the Feedback Disconnect and converting sparse signals into reusable chemical knowledge.

By explicitly linking the DMTA cycle with dynamic hierarchical TS, K-BTS transforms stochastic LLM-based molecule generation into a guided, convergent evolutionary search. Compared with classical MAB strategies, this enables more efficient exploration of chemical space, systematic closure of the DMTA loop, and the accumulation of reusable knowledge for iterative molecular design.

## B. Theoretical Intuition Analysis

We analyze the convergence properties of the K-BTS framework, formulated as a Dynamic Hierarchical Multi-Armed Bandit (D-HMAB). Unlike standard bandit settings, both the upper-level arm set (the molecular pool $\mathcal{P}_k$) and the lower-level arm set (the action space $\mathcal{A}_k$) are dynamically expanded during the optimization process.

### B.1. Assumptions

Our analysis relies on the following assumptions, which reflect common empirical properties of structure-based drug design landscapes.

**Assumption 1 (Predominant Local Smoothness).** Although activity cliffs may exist, Although activity cliffs may exist, it is empirically observed that many molecular property landscapes, including binding affinity or docking score, display a degree of local smoothness with respect to chemical similarity: structurally similar molecules often exhibit similar properties (Bajorath, 2016).We assume that for a fixed target, the docking landscape is *predominantly locally smooth* with respect to a chemical metric $d_{\mathcal{X}}$ (e.g., Tanimoto distance). That is, for most pairs of structurally similar molecules $m_1, m_2$,

$$|f(m_1, t) - f(m_2, t)| \leq L \cdot d_{\mathcal{X}}(m_1, m_2). \tag{18}$$

This assumption captures the typical behavior away from rare discontinuities.

**Assumption 2 (Stochastic Local Reachability).** Given a seed molecule $m$ and a modification action $a$, the LLM-based generation policy assigns non-zero probability to valid molecules within a local neighborhood of $m$. This ensures that refinement trajectories can explore nearby chemical regions and that no valid local transformation is permanently excluded.

**Assumption 3 (Bounded Noisy Feedback).** Observed docking improvements $\Delta y$ are bounded and corrupted by sub-Gaussian noise, consistent with the stochasticity of docking evaluations.

### B.2. Convergence Analysis

#### B.2.1. LOWER-LEVEL CONSISTENCY WITH DYNAMIC ARMS

**Lemma B.1** (Dynamic Action Selection Consistency). *Consider a fixed scaffold context $S$. Let $\mathcal{A}_{\mathrm{univ}}(S)$ denote the finite but unknown set of all chemically valid modification intents for $S$, and let $\mathcal{A}_k(S) \subseteq \mathcal{A}_{\mathrm{univ}}(S)$ be the subset discovered up to iteration $k$. Each newly discovered action is initialized with a non-informative prior $(\alpha_0, \beta_0) = (1, 1)$.*

*Conditioned on the discovered action set $\mathcal{A}_k(S)$, the lower-level policy asymptotically identifies an optimal action within $\mathcal{A}_k(S)$. Once the globally optimal action $a^* \in \mathcal{A}_{\mathrm{univ}}(S)$ enters the pool, it is selected with high probability. Moreover, the cumulative lower-level regret is sub-linear in time.*

*Proof.* The dynamic expansion of $\mathcal{A}_k$ corresponds to a bandit problem with sequentially revealed arms. Convergence follows from the following observations.

**(i) Bounded Action Universe.** Although $\mathcal{A}_k$ grows over time, it is bounded by $\mathcal{A}_{\mathrm{univ}}(S)$, which is finite due to the constrained grammar of chemically valid transformations for a fixed scaffold. Hence, $\lim_{k \to \infty} \mathcal{A}_k = \mathcal{A}_{\mathrm{univ}}$.

**(ii) Uncertainty-Driven Exploration.** Newly introduced actions are initialized with $\mathrm{Beta}(1, 1)$ priors. Under Thompson Sampling, such actions are selected with high probability whenever their posterior samples exceed those of incumbent actions. This ensures that newly discovered intents are sufficiently evaluated before being discarded.

**(iii) Rapid Elimination of Suboptimal Actions.** For suboptimal actions, posterior variance decreases as more observations are accumulated, causing their selection probability to decay rapidly as evidence against them grows.

**(iv) Regret Bound.** The total regret can be decomposed into the standard Thompson Sampling regret over the revealed arms and a constant validation cost for each newly introduced arm:

$$R_T^L \;\leq\; \sum_{a \in \mathcal{A}_{\mathrm{univ}}} \frac{\ln T}{\Delta_a} \;+\; C\,|\mathcal{A}_{\mathrm{univ}}|. \tag{19}$$

Since $|\mathcal{A}_{\mathrm{univ}}|$ is finite, the average regret $R_T^L/T \to 0$ as $T \to \infty$. $\qquad\square$

#### B.2.2. UPPER-LEVEL SELECTION BIAS TOWARD HIGH-POTENTIAL SEEDS

**Lemma B.2** (Asymptotic Bias in Upper-Level Selection). *The upper-level selection score $S(m) = \hat{\theta}_m \cdot \exp(-y_m/\tau)$ induces an asymptotic sampling bias toward molecules that jointly exhibit low docking energy and high refinement potential within the dynamically expanding pool $\mathcal{P}_k$, where $\tau$ is the sampling temperature.*

*Proof.* The selection rule combines a Boltzmann-weighted preference for low-energy molecules with Bayesian uncertainty over refinement potential. The exponential term increasingly favors molecules with lower docking scores, while the stochasticity of $\hat{\theta}_m$ ensures non-vanishing exploration of uncertain or newly generated candidates. The magnitude-aware update $\alpha_m \leftarrow \alpha_m + \lambda|\Delta y|$ causes lineages that consistently yield significant improvements to accumulate posterior mass more rapidly, biasing future selections toward high-potential regions of chemical space. $\qquad\square$

#### B.2.3. GLOBAL CONVERGENCE

**Proposition B.3** (Asymptotic Improvement Tendency of K-BTS). *Under Assumptions 1–3 and Lemmas B.1–B.2, the K-BTS framework exhibits an asymptotic improvement tendency toward high-affinity regions of the chemical space. Specifically, for any $\epsilon > 0$, there exists $K_\epsilon$ such that*

$$\mathbb{P}\Big(f(m_{\mathrm{best}}^{(k)}) - f(m^*) < \epsilon\Big) \tag{20}$$

*tends to increase for sufficiently large $k$, and approaches 1 as exploration continues.*

*Proof.* The optimization process can be interpreted as a stochastic search over a chemically smooth manifold.

**(i) Local Improvement.** By Lemma B.1, for any visited scaffold, the lower-level policy increasingly favors actions with higher expected improvement within the discovered action set. Consequently, conditional on a fixed molecular lineage, the local search process tends to yield progressively improved docking outcomes in expectation.

**(ii) Global Allocation of Search Budget.** Lemma B.2 suggests that the upper-level policy adaptively allocates more sampling probability toward molecular lineages with higher posterior potential for affinity improvement, while gradually reducing exploration of persistently low-potential branches.

**(iii) Reachability of Chemical Space.** Under Assumption 2, the induced stochastic process maintains non-zero probability of exploring all reachable regions of the chemically valid search space. Therefore, no suboptimal local basin becomes permanently absorbing with probability one.

**(iv) Asymptotic Improvement Behavior.** The sequence of best observed docking scores $\{f(m_{\text{best}}^{(k)})\}_{k \geq 1}$ is bounded below by the physical limits of binding affinity. As exploration proceeds, the combination of local improvement and continued global exploration increases the likelihood of discovering molecules closer to the optimal binding region. Consequently, the probability

$$\mathbb{P}\Big(f(m_{\text{best}}^{(k)}) - f(m^*) < \epsilon\Big) \tag{21}$$

tends to increase for sufficiently large $k$. This establishes the asymptotic improvement tendency stated in the proposition. □

## C. Method Details

### C.1. Algorithm Pseudocode

The overall optimization procedure of **K-BTS** is summarized in Algorithm 1.

### C.2. LLM Prompt Design

**System Instruction (Role Definition).** The following system prompt is fixed across all targets and iterations, defining the LLM's professional role and global constraints.

---

**System Instruction**

```
You are an expert medicinal chemist specializing in structure-based drug design (SBDD).
Your goal is to optimize a lead compound to improve its binding affinity (lowering
docking scores).
You are proficient in SMILES/SMARTS notation, bioisosteric replacement, and medicinal
chemistry principles.
Always ensure that the generated molecules are chemically valid and maintain
reasonable drug-likeness.
```

---

**Iteration-Specific Task Prompt.** At each iteration, the user prompt is instantiated from the current algorithmic state, including the selected seed molecule, local history, and the chosen knowledge action (if any). A template example is shown below.

---

**Iteration-Specific User Prompt Template**

```
### Current Lead Molecule
- SMILES: <SMILES of m_seed>
- Current Docking Score:  <Score>
- Scaffold:  <Murcko Scaffold>

### Forbidden Results (Already Tested)
The following derivatives of this scaffold have already been evaluated and must NOT be
repeated:
```

---

```
- <SMILES₁>
- <SMILES₂>
- ...

### Strategic Inspiration (From Historical Success)
- Evidence:  This strategy was previously successful for this or a similar scaffold.
- Insight:  <Description of the chemical transformation and its utility>
- Task Guidance:  <Action-level instruction derived from $a_k$>

### Task Requirements
1.  Provide exactly ONE new SMILES string.
2.  The result MUST be different from the seed and all forbidden results.
3.  Ensure chemical validity and briefly explain the rationale.

Response Format:
Rationale:  <Explanation>
New SMILES: <SMILES string>
```

## C.3. Example Prompts and LLM Outputs

We present two key iterations of K-BTS to illustrate R-group modification (Iteration 16) and Scaffold Hopping (Iteration 81) in Figure 3. Each box contains the system instruction, user task prompt, and the resulting LLM output.

### Iteration 16: R-group Modification

```
System Instruction:
You are an expert medicinal chemist specializing in structure-based drug design
(SBDD).
Your goal is to optimize a lead compound to improve its binding affinity (lowering
docking scores).
You are proficient in SMILES/SMARTS notation, bioisosteric replacement, and medicinal
chemistry principles.
Always ensure that the generated molecules are chemically valid and maintain
reasonable drug-likeness.

User Task Prompt:
### Current Lead Molecule
- SMILES: C[C@H](c1ccc(-c2ccccc2)c(F)c1)S(=O)(=O)NCC(=O)NO
- Current Docking Score:  -9.90
- Scaffold:  c1ccc(-c2ccccc2)cc1

### Forbidden Results (Already Tested)
- C[C@H](c1ccc(-c2ccccc2)c(F)c1)S(=O)(=O)NCCS(=O)(=O)NN=O
- C[C@H](c1ccc(-c2ccccc2)c(F)c1)S(=O)(=O)NCC(=O)S(=O)(=O)NN=O
- C[C@H](c1ccc(-c2ccccc2)c(F)c1)S(=O)(=O)NCC(=O)NN=O
- C[C@@H](C(=O)O)c1ccc(-c2ccccc2)c(F)c1
- C[C@H](c1ccc(-c2ccccc2)c(F)c1)S(=O)(=O)NCC(=S)NN=O

### Strategic Inspiration (From Historical Success)
- Evidence:  This strategy was previously successful for THIS scaffold.
- Insight:  Replacing group [*[C@@H](C)S(=O)(=O)NCC(=S)NN=O] with
[*[C@@H](C)S(=O)(=O)NCC(=O)S(=O)(=O)NN=O] was highly effective (Utility:  0.34).
- Task Guidance:  ACTION: Design a modification inspired by the shift toward
[*[C@@H](C)S(=O)(=O)NCC(=O)S(=O)(=O)NN=O].

### Task Requirements
1.  Provide exactly ONE new SMILES string.
2.  MANDATORY: The result MUST be different from the Seed and all Forbidden Results.
3.  Ensure chemical validity and explain your rationale.
Response Format:
Rationale:  <Your explanation>
New SMILES: <The new SMILES string>
```

**LLM Output:**
Rationale: The strategic inspiration highlights that replacing the sulfonamide-linked thioamide group (C(=S)NN=O) with a sulfonyl-linked oxamide-like group (C(=O)S(=O)(=O)NN=O) was highly effective. This suggests that introducing additional sulfonyl groups or modifying the terminal warhead to enhance hydrogen bonding or electrostatic interactions can improve binding. However, the forbidden list already includes the exact sulfonyl oxamide derivative (C[C@H](c1ccc(-c2ccccc2)c(F)c1)S(=O)(=O)NCC(=O)S(=O)(=O)NN=O). To bypass this, I propose a bioisosteric replacement of the terminal NN=O (nitrosohydrazine) with a 1,2,4-oxadiazole-3-one ring, which maintains similar hydrogen bond acceptor properties and π-conjugation but is distinct in structure. Specifically, I will replace the NCC(=O)S(=O)(=O)NN=O tail with NCC(=O)S(=O)(=O)N1N=NC1=O, forming a cyclic sulfonylurea-oxadiazolone hybrid. This retains the sulfonyl and carbonyl motifs from the inspiration while introducing a novel heterocycle not in the forbidden list, potentially improving affinity through rigidification and additional dipole interactions.
New SMILES: C[C@H](c1ccc(-c2ccccc2)c(F)c1)S(=O)(=O)NCC(=O)S(=O)(=O)N1N=NC1=O

---

## Iteration 81: Scaffold Hopping

**System Instruction:**
You are an expert medicinal chemist specializing in structure-based drug design (SBDD).
Your goal is to optimize a lead compound to improve its binding affinity (lowering docking scores).
You are proficient in SMILES/SMARTS notation, bioisosteric replacement, and medicinal chemistry principles.
Always ensure that the generated molecules are chemically valid and maintain reasonable drug-likeness.
**User Task Prompt:**

### Current Lead Molecule
- SMILES: O=S(=O)(Cc1ccc(-c2ccccc2)cc1)Nc1noc2ccccc12
- Current Docking Score: -10.40
- Scaffold: O=S(=O)(Cc1ccc(-c2ccccc2)cc1)Nc1noc2ccccc12

### Strategic Inspiration (From Historical Success)
- Evidence: This strategy was previously successful for THIS scaffold.
- Insight: Transitioning from core [O=S(=O)(Cc1ccc(-c2ccccc2)cc1)Nc1noc2ccccc12] to [O=C1N=NN1S(=O)(=O)C(=O)CNS(=O)(=O)Cc1ccc(-c2ccccc2)cc1] improved binding (Utility: 0.25).
- Task Guidance: ACTION: Perform a core replacement or structural reorganization inspired by this scaffold hop.

### Task Requirements
1. Provide exactly ONE new SMILES string.
2. MANDATORY: The result MUST be different from the Seed and all Forbidden Results.
3. Ensure chemical validity and explain your rationale.

Response Format:
Rationale: <Your explanation>
New SMILES: <The new SMILES string>
**LLM Output:**
Rationale: The inspiration suggests a scaffold hop where the original sulfonamide-linked biphenyl-isoxazole system is replaced by a more complex, fused heterocyclic core containing additional sulfonyl and carbonyl groups. Specifically, the original core 'O=S(=O)(Cc1ccc(-c2ccccc2)cc1)Nc1noc2ccccc12' was transformed into a structure with a central 1,2,4-triazole-3-one ring ('O=C1N=NN1') linked to sulfonyl and carbonyl groups, extending the pharmacophore. To adapt this while avoiding forbidden structures, I designed a new molecule that: 1. Replaces the isoxazole ring with a 1,2,4-triazole-3-one moiety (inspired by 'O=C1N=NN1'), which can enhance hydrogen bonding and dipole interactions. 2. Retains the biphenyl methylene

```
linker ('Cc1ccc(-c2ccccc2)cc1') to preserve hydrophobic contacts.  3.  Introduces a
sulfonylcarbamoyl bridge ('S(=O)(=O)C(=O)NC') between the triazole and the biphenyl
moiety, mimicking the inspiration's sulfonyl/carbonyl spacing but with an amide
instead of an ether linkage to ensure novelty.  4.  Ensures chemical validity by
maintaining balanced polarity and avoiding unstable combinations (e.g., no reactive
groups).  This redesign aims to improve binding affinity by adding complementary
hydrogen bond acceptors/donors and rigidifying the core, potentially lowering the
docking score while adhering to the scaffold-hop strategy.
New SMILES: O=C1N=NN1S(=O)(=O)C(=O)NCc1ccc(-c2ccccc2)cc1
```

## D. Detailed Experimental Settings

### D.1. Hyperparameter Settings

**For our method**, the threshold used in the warm-start phase for preference comparison, i.e., $y_i - y_j > \epsilon$, was set to $\epsilon = 0.5$. In the upper-level Thompson sampling procedure, the Boltzmann temperature $T$ was selected from the candidate set $\{0.5, 2, 10\}$, among which $T = 2$ yielded the best empirical performance and was therefore adopted.

For the lower-level Thompson sampling, a confidence threshold $\tau$ was introduced to determine whether to trigger free exploration. We evaluated $\tau \in \{0.1, 0.3, 0.5\}$ and finally chose $\tau = 0.3$ based on validation performance. In the experience inheritance module, the inheritance constant $\gamma$ was tuned over $\{0.3, 0.4, 0.5\}$, with $\gamma = 0.5$ providing the most stable optimization behavior. During the Bayesian update in the upper-level controller, the scaling factor $\lambda$ was selected from $\{2, 3, 5\}$, and $\lambda = 3$ was used in all experiments.

For LLM-based reasoning and molecular modification, we employed *DeepSeek-V3.2-Chat* as the backbone model, with the sampling temperature set to $0.7$ to balance diversity and stability.

For docking-based evaluation, `rdkit-2024.9.6` was used to construct 3D molecular conformations. The ETKDGv3 parameter set was adopted with a fixed random seed of $1$. After conformer generation, all molecular structures were further optimized using a deterministic MMFF force field. Docking scores were then computed using `Smina`, with the following parameters: `autobox_add = 1, seed = 1234, and exhaustiveness = 32`. All docking experiments were conducted using 30 CPU cores.

**For the baselines**, we compare K-BTS against three categories of representative SBDD methods. (1) **Structure-based generative models**, including the voxel-based VAE **liGAN** (Ragoza et al., 2022) and the 3D atomic-level autoregressive models **AR** (Luo et al., 2021) and **Pocket2Mol** (Peng et al., 2022). (2) **Diffusion models**, comprising **TargetDiff** (Guan et al., 2023a) and the state-of-the-art **ALIDiff** (Gu et al., 2024). (3) **LLM-based models**, including **TamGen** (Wu et al., 2024), a GPT-based chemical language model for conditional de novo generation, as well as the iterative refinement approaches **LMLF** (Brahmavar et al., 2024) and **ELILLM** (Hu et al., 2026). Except for LMLF, the reported results of all baselines are taken directly from the original **ELILLM** implementation to ensure faithful reproduction.

For **LMLF**, we re-implement the method using the **same DeepSeek-V3.2 API** as K-BTS to ensure a fair comparison. To guarantee that LMLF can successfully generate 100 ligands under our experimental setting, we set the *docking threshold* to $-6.5$ and the *threshold increment frequency* to $50$, meaning that the threshold is decreased by 1 every 50 iterations.

For a fair comparison, **K-BTS**, **LMLF**, and **ELILLM** are evaluated under both **-rand** and **-diff** settings, using the same initial molecules as ELILLM.

### D.2. System and Software Configuration

All experiments were conducted on Linux-based servers with CPU-only computation. Due to different computational characteristics, the **-rand** and **-diff** variants were executed on separate hardware environments.

Specifically, the **-rand** experiments were performed on a server equipped with an **Intel(R) Xeon(R) Silver 4310 CPU @ 2.10GHz**, while the **-diff** experiments were conducted on a server equipped with an **Intel(R) Xeon(R) Gold 6430** processor. Both processors belong to Intel Xeon Scalable families and support multi-core parallel execution. We note that the difference in hardware configurations primarily affects overall runtime efficiency, while the core optimization logic, scoring functions, and decision-making processes remain identical across experimental settings.

For the software environment, molecular representation, manipulation, and validation were implemented using **RDKit 2024.09.6**. Molecular docking and binding affinity estimation were carried out using **Smina 2020.12.10**. All large language model interactions were handled by **DeepSeek v3.2**, specifically the `deepseek-chat` model, which was used uniformly across all experiments.

### D.3. Code and Reproducibility

To ensure reproducibility, we include all code and scripts necessary in an anonymous online repository at https://github.com/hxnhxn/K-BTS (or Supplementary materials). All raw results, including molecule SMILES, docking scores, auxiliary metrics, generation prompts, and docking poses, are provided in the results/ directory, enabling full reproduction of our experiments.

### D.4. The Crossdocked2020 dataset

We evaluated our method on the CrossDocked2020 dataset (Francoeur et al., 2020), which contains over 22 million cross-docked ligand poses across diverse protein targets. CrossDocked2020 presents docking poses into non-cognate receptor structures, making it more appropriate for structure-based drug design tasks. Its large scale, pose diversity, and cluster-split test sets provide a robust benchmark for evaluating both affinity prediction and ligand generation methods. We primarily evaluate on the test set, which offers a large and diverse set of realistic docking poses to ensure comprehensive and robust assessment across a wide range of protein-ligand interactions. Recent SBDD models, including TargetDIFF (Guan et al., 2023a), ALIDIFF (Gu et al., 2024), TamGen (Wu et al., 2024), ELILLM (Hu et al., 2026) and others, also utilize CrossDocked2020, facilitating fair comparison and enhancing community relevance.

### D.5. Evaluation Metrics Details

We adopted multiple widely used molecular evaluation metrics to assess the quality of generated ligands: Vina Docking Score, QED, Synthetic Accessibility (SA), Molecular Diversity, and Tanimoto Similarity. Among these, **Vina Docking Score** serves as the *primary objective*, guiding the molecule generation and optimization in the latent space. The remaining metrics act as *auxiliary indicators* to provide complementary assessments of drug-likeness, synthesizability and chemical diversity.

**Vina Docking Score** (Trott & Olson, 2010) quantifies the predicted binding affinity between a ligand and the protein pocket. We use Smina (Koes et al., 2013) to calculate docking scores, selecting the lowest score among all generated poses as the final result.

**QED** (Bickerton et al., 2012) reflects how "drug-like" a molecule is. Following RDKit's implementation, it is computed by aggregating desirability functions of eight key molecular properties: molecular weight (MW), octanol-water partition coefficient (ALOGP), number of hydrogen bond acceptors (HBA), number of hydrogen bond donors (HBD), polar surface area (PSA), number of rotatable bonds (ROTB), number of aromatic rings (AROM), and the number of structural alerts (ALERTS). The QED score is defined as the weighted geometric mean of these desirability functions:

$$\text{QED}(m) = \exp\left(\frac{\sum_i w_i \log d_i(m)}{\sum_i w_i}\right) \tag{22}$$

where $d_i(m)$ is the desirability of property $i$ for molecule $m$, and $w_i$ is the corresponding weight.

**SA Score** (Ertl & Schuffenhauer, 2009) estimates synthetic accessibility based on molecular complexity and the presence of uncommon substructures. We use RDKit's implementation.

**Diversity** is measured by the average pairwise Tanimoto distance between RDKit fingerprints of generated molecules:

$$\text{Diversity} = \frac{2}{n(n-1)} \sum_{i<j} (1 - \text{Tanimoto}(f_i, f_j)) \tag{23}$$

where $f_i$ and $f_j$ denote the molecular fingerprints.

**Tanimoto Similarity** (Bajusz et al., 2015) to known active ligands is used to evaluate how novel or distinct the generated molecules are relative to existing bioactive compounds.

*Table 4.* Holm–Bonferroni corrected p-values for paired Wilcoxon tests comparing K-BTS-diff with strong baselines across 100 targets under different Top-$K$ settings.

| Baseline Method | Top-1 | Top-5 | Top-10 | Top-20 |
|---|---|---|---|---|
| ELILLM-diff | 2.01e-7 | 5.19e-12 | 5.86e-14 | 3.41e-15 |
| LMLF-diff | 3.65e-15 | 8.03e-18 | 7.79e-18 | 7.79e-18 |
| ALIDiff | 1.09e-16 | 8.03e-18 | 7.79e-18 | 7.79e-18 |
| TargetDiff | 2.01e-12 | 1.85e-15 | 3.19e-16 | 3.05e-17 |

# E. Supplementary Experiments

## E.1. Statistical Significance Analysis

**Evaluation Objective.** We aim to rigorously verify whether **K-BTS-diff** consistently outperforms several strong baselines (ELILLM-diff, LMLF-diff, ALIDiff, and TargetDiff) in terms of binding affinity across different Top-$K$ evaluation settings ($K \in \{1, 5, 10, 20\}$). All evaluations are conducted on the same set of 100 protein targets from the CrossDocked2020 benchmark, enabling paired statistical comparisons.

For each target and each Top-$K$ setting, we compute the *mean docking score* across the top-$K$ ranked candidate molecules for each method, resulting in one representative value per target. This yields 100 paired observations for each (baseline, Top-$K$) comparison.

**Paired Hypothesis Testing.** Let $x_i^{(K)}$ and $y_i^{(K)}$ denote the mean docking scores of **K-BTS-diff** and a baseline method, respectively, on target $i$ under the Top-$K$ setting. Lower docking scores indicate better binding affinity. We define the paired differences as:

$$d_i^{(K)} = x_i^{(K)} - y_i^{(K)}. \tag{24}$$

For each Top-$K$ setting, we employ a one-sided Wilcoxon signed-rank test with the following hypotheses:

$$H_0 : \text{median}(d_i^{(K)}) \geq 0 \tag{25}$$

$$H_1 : \text{median}(d_i^{(K)}) < 0 \tag{26}$$

**Multiple Hypothesis Correction.** For each Top-$K$ setting, we apply the Holm–Bonferroni procedure across baseline comparisons to control the family-wise error rate (FWER) at $\alpha = 0.05$. Corrections are performed independently for each Top-$K$, as these metrics reflect different evaluation criteria.

**Statistical Test Results.** Table 4 reports the Holm–Bonferroni corrected p-values of one-sided paired Wilcoxon signed-rank tests comparing **K-BTS-diff** with each baseline across different Top-$K$ settings.

## E.2. Visualization of Paired Performance Differences

To complement the numerical significance tests, we visualize the distribution of paired docking score differences across targets. For each baseline, we compute:

$$\Delta_i = y_i - x_i, \tag{27}$$

where $y_i$ and $x_i$ denote the docking scores of the baseline and **K-BTS-diff** on target $i$, respectively. Positive values indicate that **K-BTS-diff** achieves lower (better) docking scores.

Figure 5 presents boxplots of $\Delta_i$ for Top-1, Top-5, Top-10, and Top-20 settings. The box shows the interquartile range (middle 50% of targets), the line inside the box indicates the median, whiskers represent the range excluding extreme outliers, and points beyond whiskers correspond to rare targets where the baseline outperforms **K-BTS-diff**. Across all Top-$K$ settings, the boxes are predominantly above zero, corroborating the results of one-sided paired Wilcoxon tests with Holm–Bonferroni correction: **K-BTS-diff** consistently outperforms all considered baselines.

## E.3. Supplementary Analysis: Iterative Evolution Curves

To complement the main text, where only random baseline curves were shown, we provide a detailed analysis of iterative optimization trends across 100 targets. Both **K-BTS-diff** and baseline methods are evaluated in batches of 10 molecules to

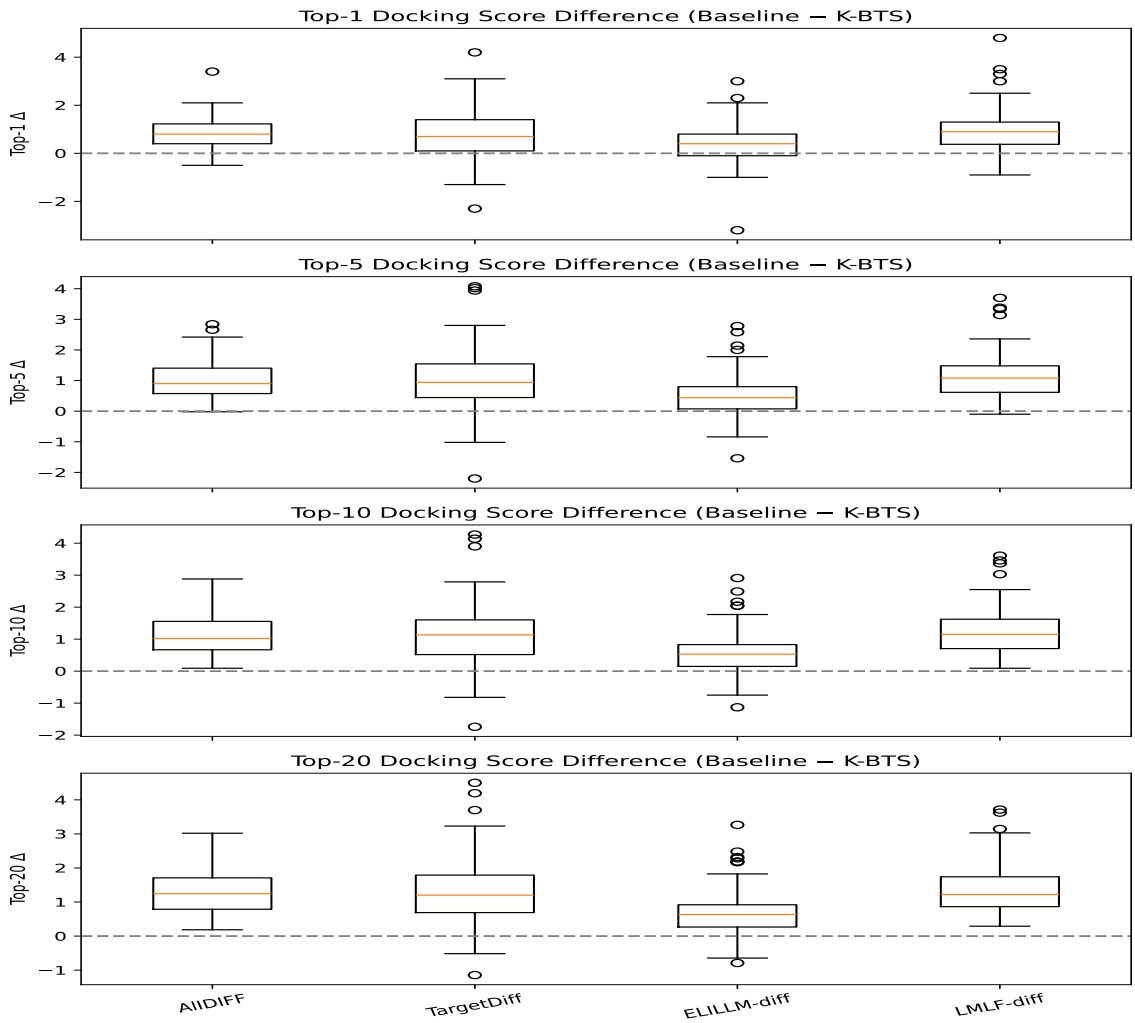

*Figure 5.* Paired docking score differences between **K-BTS-diff** and each baseline across 100 targets. The dashed horizontal line at zero indicates equal performance; positive values correspond to targets where **K-BTS-diff** outperforms the baseline.

capture instantaneous trends in docking score optimization and structural changes. These trends are visualized in Figure 6, where (a) shows average docking scores and (b) shows structural drift measured by maximum Tanimoto similarity to the initial set $\mathcal{D}_0$.

### E.4. Additional Ablation Study

We report the same ablation experiments under the **-diff** setting in Table 5. To further investigate the role of warm-start initialization, we additionally present the per-8-iteration average docking score trajectories of all variants under the **-rand** setting, focusing on the early-stage optimization behavior across 10 targets up to iteration 40. From the curves in Figure 7, K-BTS-**rand** exhibits a markedly faster performance improvement and smaller variance in the early iterations compared to the **wo_warmstart** variant, highlighting the benefit of warm-start initialization. Notably, the **wo_warmstart** and **wo_knowledge** variants show highly similar early-stage trajectories, as the former relies almost entirely on unguided LLM exploration in the absence of prior knowledge, which closely resembles the behavior of the latter.

The **wo_upper** variant consistently yields the worst performance throughout the optimization process, underscoring the necessity of upper-level seed selection. Without an effective mechanism to prioritize high-potential molecular lineages, the lower-level knowledge becomes overly dispersed, leading to inefficient exploration.

We further observe that the **wo_lower** variant exhibits pronounced fluctuations around iteration 16 and converges to performance comparable to **wo_knowledge** after iteration 24. This behavior can be attributed to the broad and heterogeneous

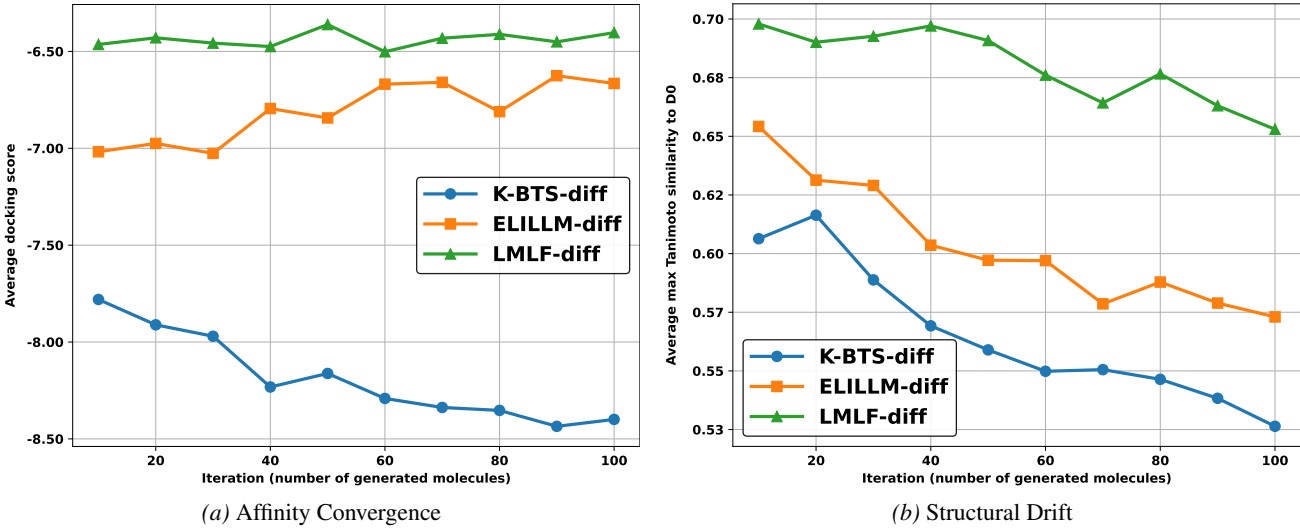

*(a) Affinity Convergence*          *(b) Structural Drift*

*Figure 6.* -diff setting: Iterative evolution analysis across 100 targets. (a) Average docking score and (b) Average maximum Tanimoto similarity to $\mathcal{D}_0$, computed independently for each batch of 10 molecules to capture instantaneous optimization trends.

*Table 5.* Ablation study of K-BTS components on the first 10 targets of the test set. All variants are initialized with (**ALIDIFF**).

| Methods | Top1 Dock ↓ | | Top20 Dock↓ | | *Diversity* | |
|---|---|---|---|---|---|---|
| | Avg. | Med. | Avg. | Med. | Avg. | Med. |
| K-BTS-diff | **-10.26** | **-10.26** | **-9.52** | **-9.47** | 0.60 | 0.63 |
| w/o upper | -9.12 | -9.12 | -8.16 | -8.09 | 0.77 | 0.75 |
| w/o lower | -9.88 | -9.88 | -9.03 | -8.92 | 0.68 | 0.70 |
| w/o knowledge | -9.80 | -9.80 | -9.16 | -9.12 | 0.59 | 0.57 |
| w/o warm-start | -9.96 | -9.96 | -9.29 | -9.23 | 0.67 | 0.72 |
| ALIDIFF | -9.10 | -9.10 | -8.17 | -8.04 | 0.75 | 0.72 |

nature of the warm-start action knowledge: in early iterations, randomly retrieved actions may be suboptimal or even detrimental. As optimization proceeds, more focused and empirically validated actions are progressively extracted, allowing the performance of **wo_lower** to gradually recover. This observation further highlights the importance of the proposed lower-level knowledge retrieval mechanism.

### E.5. Detailed Optimization Trajectory for Target 4PXZ

This section provides a step-by-step analysis of the optimization trajectory for target 4PXZ, as visualized in Fig. 3. The trajectory demonstrates how K-BTS navigates complex chemical spaces by balancing exploration and exploitation through its bi-level architecture and knowledge-guided reasoning.

E.5.1. STEP I: PRIOR-DRIVEN FUNCTIONAL GROUP SUBSTITUTION (ITERATION 1)

- **Current State:** Root molecule (SMILES: `C[C@@H](C(=O)O)c1ccc(-c2ccccc2)c(F)c1`)

- **Metrics:** Vina Score: -9.4 — QED: 0.89 — SA Score: 0.78

- **Retrieved Insight:** Replacing the carboxyl group `[C(=O)O]` with a sulfonamide-linked nitroacetamide group `[S(=O)(=O)NCC(=O)NO]` (Utility: 0.54).

- **Reasoning:** The LLM identifies that replacing the carboxylate with a sulfonamide tether may improve binding affinity through additional hydrogen bonding and electrostatic interactions while preserving the hydrophobic biphenyl scaffold.

- **Evolutionary Outcome: Step 1** (`C[C@H](c1ccc(-c2ccccc2)c(F)c1)S(=O)(=O)NCC(=O)NO`)

- **Outcome Metrics:** Vina Score: -9.9 — QED: 0.55 — SA Score: 0.79

E.5.2. STEP II: CONSTRAINT-INDUCED INNOVATION UNDER $\mathcal{H}_m$ (ITERATION 16)

- **Current State:** Step 1 lead molecule (Increasing pressure from the Forbidden List $\mathcal{H}_m$)

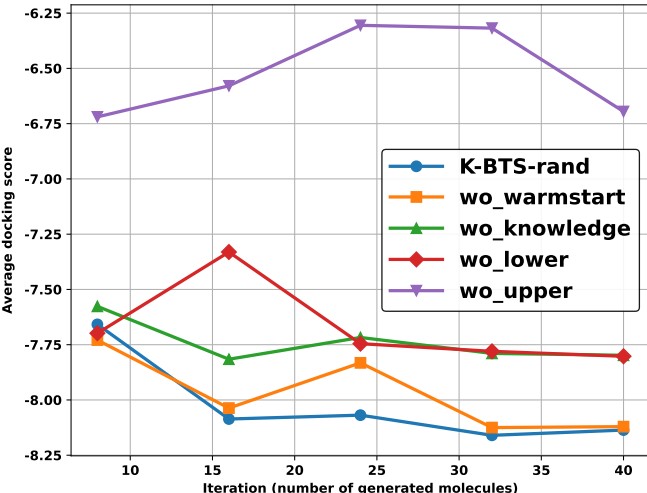

*Figure 7.* Iterative evolution analysis of the **-rand** variant across 10 targets (8 iterations per step, up to iteration 40).

- **Metrics:** Vina Score: -9.9 — QED: 0.55 — SA Score: 0.79

- **Retrieved Insight:** Modifying the terminal warhead with a sulfonyl-linked oxamide-like group (Utility: 0.34).

- **Reasoning:** To bypass $\mathcal{H}_m$, the LLM executes a bioisosteric replacement of the terminal nitrosohydrazine with a **1,2,4-oxadiazole-3-one ring**. This cyclization increases structural rigidity and introduces novel dipole interactions, leading to a significant jump in binding affinity.

- **Evolutionary Outcome: Step 2** (C[C@H](c1ccc(-c2ccccc2)c(F)c1)S(=O)(=O)NCC(=O)S(=O)(=O)N1N=NC1=O)

- **Outcome Metrics:** Vina Score: -11.2 — QED: 0.68 — SA Score: 0.70

### E.5.3. STEP III: STRATEGIC REGRESSION AND MANIFOLD TRANSITION (ITERATION 31)

- **Current State:** Step 2 molecule (Local manifold exhausted)

- **Metrics:** Vina Score: -11.2 — QED: 0.68 — SA Score: 0.70

- **Retrieved Insight:** Executing a scaffold-hopping action toward a benzoxazole sulfonamide core (Utility: -0.44).

- **Reasoning:** Facing a structural trap in $\mathcal{H}_m$, the LLM demonstrates strategic resilience by intentionally accepting a score regression. It reorganizes the linker into a direct sulfonamide connection between the benzoxazole and the biphenyl side chain, successfully migrating to a new conformational territory.

- **Evolutionary Outcome: Step 3** (O=S(=O)(Cc1ccc(-c2ccccc2)cc1)Nc1noc2ccccc12)

- **Outcome Metrics:** Vina Score: -10.4 — QED: 0.57 — SA Score: 0.88

### E.5.4. STEP IV: RETRIEVAL-INSPIRED RECOMBINATION AND GLOBAL CONVERGENCE (ITERATION 81)

- **Current State:** Step 3 molecule (Exploration branch re-activated via Thompson Sampling)

- **Metrics:** Vina Score: -10.4 — QED: 0.57 — SA Score: 0.88

- **Retrieved Insight:** Transitioning from the Step 3 core to a complex fused heterocyclic system containing a **1,2,4-triazole-3-one** ring and additional sulfonyl/carbonyl groups (Utility: 0.25).

- **Reasoning:** The LLM recognizes that the $1, 2, 4$-triazole-3-one moiety in the retrieved insight can enhance hydrogen bonding. It grafts this moiety onto the Step 3 core and introduces a **sulfonylcarbamoyl bridge** (S(=O)(=O)C(=O)NC) to extend the pharmacophore. This recombination mimics the structural spacing of the retrieved successful case while ensuring novelty through an amide linkage to avoid $\mathcal{H}_m$.

- **Evolutionary Outcome: Step 4** (O=C1N=NN1S(=O)(=O)C(=O)NCc1ccc(-c2ccccc2)cc1)

- **Outcome Metrics:** Vina Score: -11.4 — QED: 0.92 — SA Score: 0.80

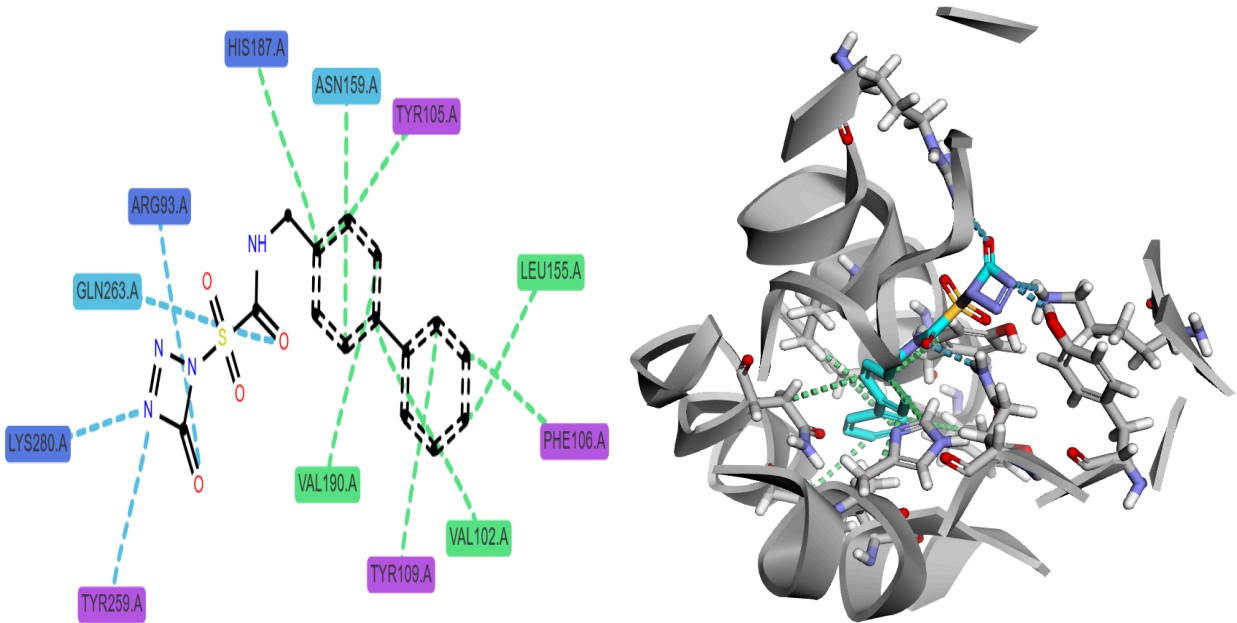

*Figure 8.* Visualization of molecular interactions for the final 4PXZ ligand. **Green** highlights hydrophobic contacts, and **blue** indicates hydrogen bonds. Left: 2D interaction summary from ProLIF. Right: 3D binding pose with interaction mapping.

### E.6. Interaction Visualization of 4PXZ Case Study

We analyze the protein–ligand interactions of the compound discovered for target 4PXZ using ProLIF (Bouysset & Fiorucci, 2021), highlighting hydrophobic contacts in green and hydrogen bonds in blue. The molecule has the following SMILES:

O=C1N=NN1S(=O)(=O)C(=O)NCc1ccc(-c2ccccc2)cc1

As shown in Figure 8, the ProLIF-based 2D interaction map (left) provides a concise overview of all contacts between the ligand and the binding pocket, while the 3D view (right) offers spatial context, showing how hydrophobic and hydrogen-bond interactions stabilize the ligand. These visualizations complement the iterative optimization narrative described in Sec. 4.5, highlighting how K-BTS progressively guides the molecule into a chemically favorable binding configuration.

### E.7. Forest Visualization

Figures 9–13 visualize the evolution forest naturally induced by K-BTS during the **-diff** optimization process on target 1E8H. Each node corresponds to a generated molecule, and node color encodes the *upper-level Boltzmann selection weight*, computed from its docking score. Darker nodes indicate lower docking energies and thus higher selection probability under the Boltzmann weighting scheme.

Overall, the forest structure reveals a characteristic *balanced exploration–exploitation* behavior emerging from K-BTS, without any explicit tree-pruning or heuristic branching rules. Instead, the structure arises organically from the interaction between uncertainty-aware seed selection and knowledge-guided action refinement.

In the early stage (Fig. 9), K-BTS performs broad exploration across multiple root lineages. During these initial iterations, the sampling budget is distributed among diverse molecular branches, enabling rapid elimination of lineages with weak short-term improvement signals.

Around iteration 10, a molecule associated with node 10 attains a relatively high Boltzmann weight due to its improved docking score, as reflected by its darker color. However, the majority of subsequent exploration concentrates on node 3, which belongs to the lineage rooted at **root4**. This behavior arises because node 3 accumulates a substantially larger $\alpha$ value in its upper-level Beta posterior, indicating stronger uncertainty-aware refinement potential. As a result, despite node 10 having a favorable instantaneous score, node 3 is sampled with higher probability under the Thompson-style upper-level selection, leading to sustained exploration along this lineage.

Evolution Forest (Target 44) Steps 1-20

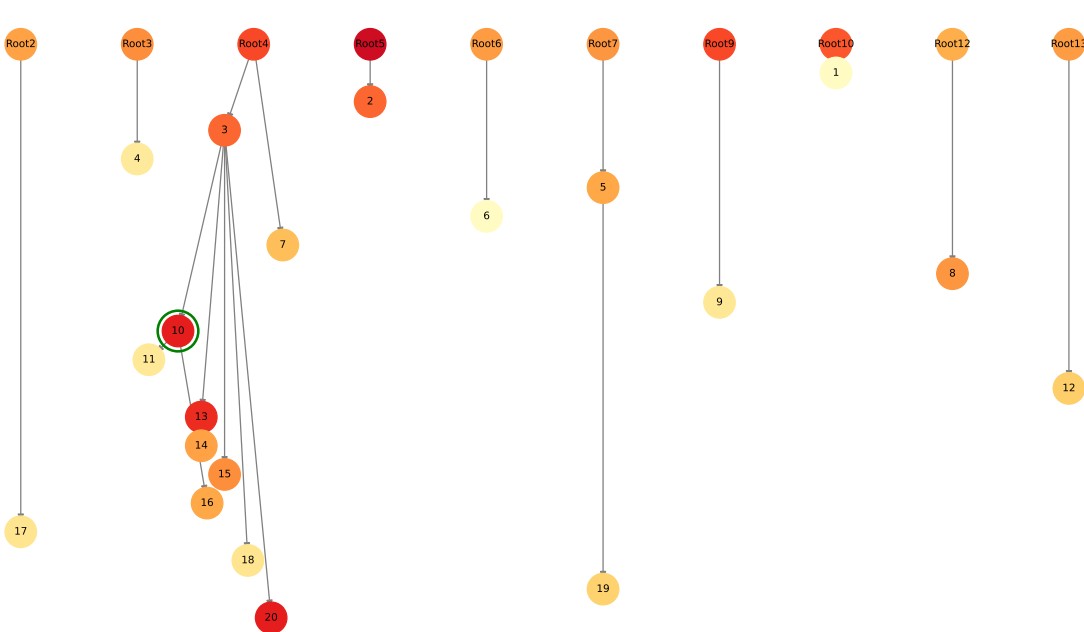

*Figure 9.* Evolution forest for steps 1-20

Evolution Forest (Target 44) Steps 21-40

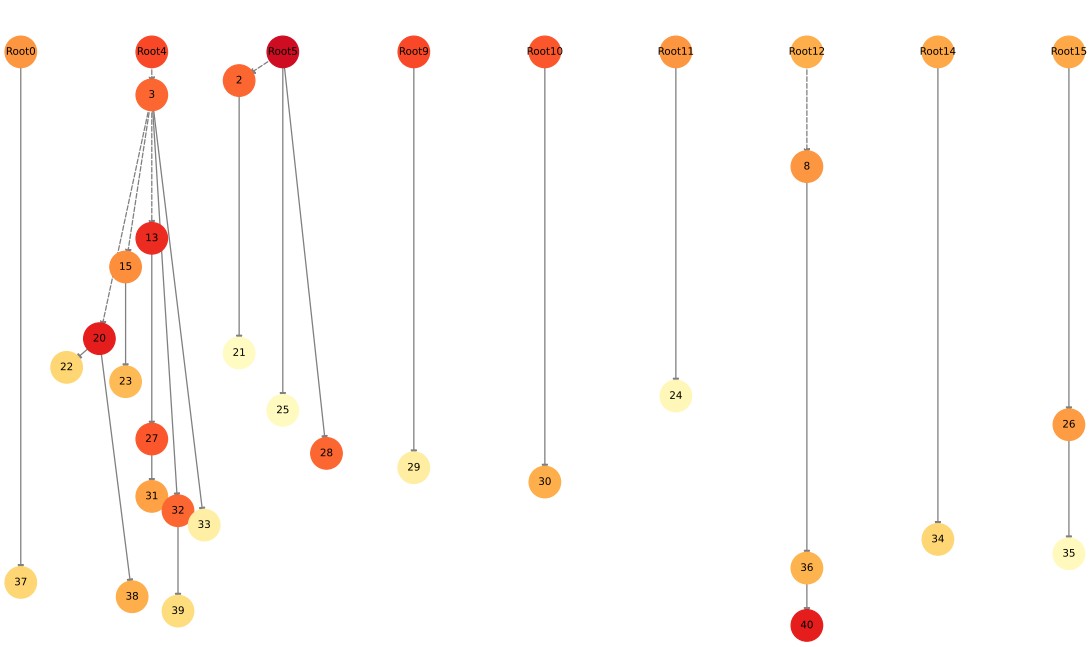

*Figure 10.* Evolution forest for steps 21-40

Evolution Forest (Target 44) Steps 41-60

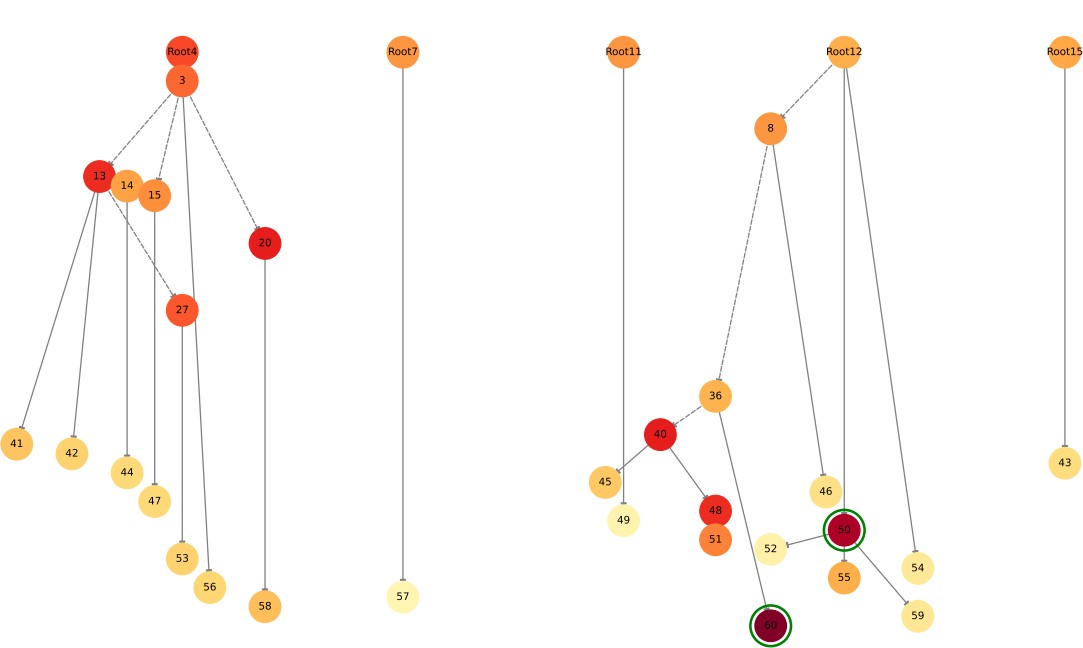

*Figure 11.* Evolution forest for steps 41-60

Evolution Forest (Target 44) Steps 61-80

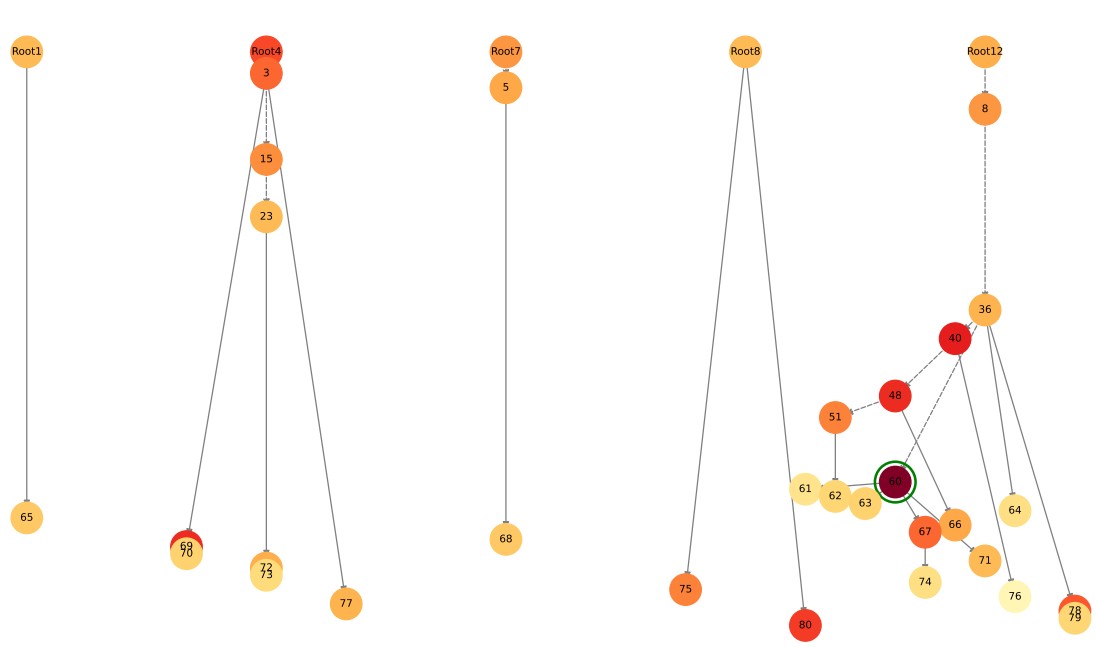

*Figure 12.* Evolution forest for steps 61-80

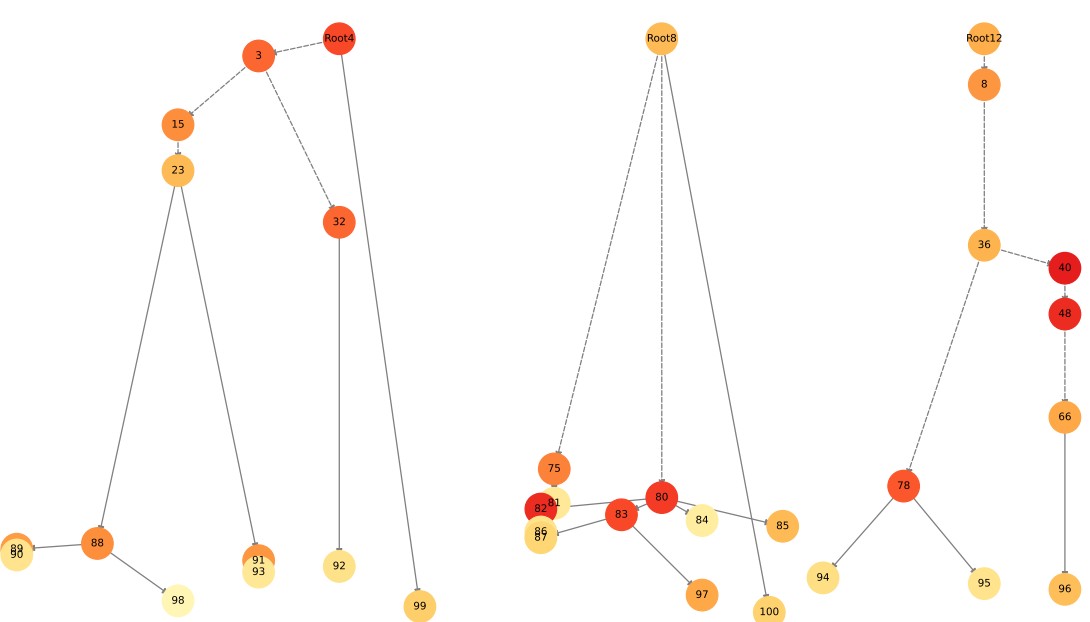

Evolution Forest (Target 44) Steps 81-100

*Figure 13.* Evolution forest for steps 81-100

As shown in Fig. 10, the search remains concentrated on the `root3` lineage, reflecting exploitation of its perceived potential. However, as local improvements become harder to obtain and the marginal gains diminish, K-BTS gradually reallocates exploration effort. During this phase, a promising molecule (node 40) with a comparatively low docking score is discovered under the `root12` lineage.

Consequently, in Fig. 11, the exploration focus shifts toward `root12`. Within this lineage, K-BTS identifies several high-quality candidates (e.g., nodes 50 and 60), reinforcing the posterior belief in the refinement potential of `root12`. This behavior persists into Fig. 12, where exploitation dominates and the search depth within the `root12` subtree increases.

Finally, Fig. 13 illustrates a renewed diversification phase. As the refinement potential of the `root12` lineage becomes saturated, K-BTS redistributes its sampling budget across multiple lineages. This multi-branch exploration enables the algorithm to escape local optima and continue searching for alternative high-affinity regions in chemical space.

Taken together, these visualizations demonstrate that K-BTS naturally induces a dynamic forest structure that alternates between focused exploitation and strategic exploration. Importantly, this behavior is not hard-coded but emerges from the bi-level Thompson Sampling mechanism and the explicit modeling of refinement potential, highlighting the interpretability and adaptivity of the proposed framework.

---

**Algorithm 1** K-BTS: Knowledge-driven Bi-level Thompson Sampling

---

**Require:** Initial molecule set $\mathcal{D}_0$, docking function $f$, target pocket $t$, LLM $\mathcal{L}$, threshold $\tau$, temperature $T$, inheritance factor $\gamma$

**Ensure:** Optimized molecular pool $\mathcal{P}$ and knowledge base $\mathcal{K}$

1: Initialize molecular pool $\mathcal{P}$ from $\mathcal{D}_0$
2: Initialize knowledge base $\mathcal{K}$ via warm-start intra-scaffold permutations (Sec. 3.2)
3: **for** iteration $k = 1, 2, \ldots, K$ **do**
4:  /* **Upper-level Thompson Sampling: Seed Selection** */
5:  **for** each $m \in \mathcal{P}$ **do**
6:    Sample $\hat{\theta}_m \sim \text{Beta}(\alpha_m, \beta_m)$
7:    Compute $S(m) = \hat{\theta}_m \cdot \exp(-y_m/T)$
8:  **end for**
9:  Select seed molecule:

$$m_{\text{seed}} = \arg\max_{m \in \mathcal{P}} S(m)$$

10:  /* **Lower-level Thompson Sampling: Action Retrieval** */
11:  Identify most similar scaffold:

$$s^* = \arg\max_s \text{Sim}(\text{MS}(m_{\text{seed}}), s)$$

12:  **if** $\text{Sim}(\text{MS}(m_{\text{seed}}), s^*) \geq \tau$ **then**
13:    **for** each $a \in \mathcal{A}_{s^*}$ **do**
14:      Sample $\hat{\theta}_a \sim \text{Beta}(\alpha_a, \beta_a)$
15:      Compute $U(a) = \hat{\theta}_a \bar{R}_{pos} + (1 - \hat{\theta}_a)\bar{R}_{neg}$
16:    **end for**
17:    Select action:

$$a_k = \arg\max_{a \in \mathcal{A}_{s^*}} U(a)$$

18:  **else**
19:    $a_k \leftarrow$ None {Free LLM exploration}
20:  **end if**
21:  /* **Knowledge-guided Generation** */
22:  Generate candidate using LLM:

$$Rationale, m_{\text{new}} = \mathcal{L}(\text{Prompt}(m_{\text{seed}}, a_k, \mathcal{H}_m))$$

23:  Evaluate docking score $y_{\text{new}} = f(m_{\text{new}}, t)$
24:  Compute affinity improvement $\Delta y = y_{\text{seed}} - y_{\text{new}}$
25:  /* **Pool Integration** */
26:  Add $m_{\text{new}}$ to $\mathcal{P}$ with inherited parameters calculated by Eq. 12(input inheritance factor $\gamma$)
27:  /* **Bidirectional Knowledge Extraction** */
28:  Extract actual actions $\mathcal{A}_{\text{new}} = \{(a, r, w)\}$ applied from $m_{\text{seed}}$ to $m_{\text{new}}$ and $m_{\text{new}}$ to $m_{\text{seed}}$
29:  For all actions in $\mathcal{A}_{\text{new}}$, update their weights using Eq. 13 and add them to the knowledge base $\mathcal{K}$
30:  /* **Bi-level Bayesian Update** */
31:  Update seed molecule parameters using Eq. 14
32:  **if** $a_k \neq$ None **then**
33:    Update action parameters using Eq. 15
34:  **end if**
35: **end for**
36: **return** $\mathcal{P}, \mathcal{K}$

---

