# OpenReview forum: "Steering Large Language Models through the DMTA Cycle: Structure-Based Drug Design via Knowledge-Driven Bi-Level Thompson Sampling"
_ICML.cc/2026/Conference — ICML 2026 regular_

### Official Review · Reviewer_g2sr · 2026-03-04

**Soundness:** 3
**Presentation:** 3
**Significance:** 2
**Originality:** 3
**Overall Recommendation:** 5
**Confidence:** 4

**Summary:**

This paper introduces K-BTS, a framework that enables LLMs to effectively execute the iterative Design-Make-Test-Analyze (DMTA) cycle for structure-based drug design. The authors identify three core disconnects in current approaches: a lack of principled exploration (Policy), an ambiguous mapping from language to chemical changes (Translation), and the inability to attribute docking scores to specific modifications (Feedback). To resolve these issues, the propposed method models the drug design process as a Dynamic Hierarchical Multi-Armed Bandit using a bi-level Thompson Sampling architecture. At the upper level, the system prioritizes promising seed molecules using Beta-Bernoulli updates, while the lower level retrieves explicit, scaffold-specific chemical rules from a dynamic knowledge base to constrain the LLM's generation step. Docking scores from the newly generated molecules are then used to update both the molecular pool and the knowledge base. The authors provide empirical evaluation of the proposed method on 100 protein targets from the CrossDocked2020 benchmark using DeepSeek-V3.2, the diffusion-initialized variant (K-BTS-diff) achieves a Top-1 average Vina docking score of -10.23, and consistently outperforming recent structure-based, diffusion, and LLM-driven baselines.

**Compliance With Llm Reviewing Policy:**

Affirmed.

**Final Justification:**

I think this paper should be accepted for all of the reasons indicated in my initial review. I initially gave a score of 4, and have now updated my score to 5 to reflect that the author's rebuttal addressed all of my questions and concerns.

**Key Questions For Authors:**

**Q1:** How robust is the method to LLM hallucinations? In my reading of Appendix C.3, the LLM sometimes generates chemically sophisticated rationales -- but what fraction of LLM-proposed modifications are chemically invalid or lead to worse docking scores? An error analysis would be valuable.

**Q2:** The paper includes computational time analysis (Table 3, Appendix E.1), showing LLM calls dominate runtime (~65 hours for -rand). Have you considered caching or amortizing LLM calls across similar scaffolds to reduce this cost?

**Q3:** How would K-BTS extend to multi-objective optimization (e.g., binding + synthesizability + ADMET)? The D-HMAB reward function currently uses only docking score improvement. Could Pareto-aware Thompson Sampling or scalarized multi-objective rewards fit naturally into this framework?

**Limitations:**

yes

**Strengths And Weaknesses:**

### Soundness
The D-HMAB formalization is mathematically clean, with well-defined state spaces, action spaces, reward functions, and transition dynamics. I found the bi-level decomposition well-motivated: the upper level maintains Beta distributions over seed molecules while the lower level leverages a knowledge base of scaffold-action pairs with their own Beta parameters. The knowledge-driven action retrieval is a clever design. Rather than relying solely on the LLM's internal priors, the system first retrieves a historical action ($a_k$) based on scaffold similarity to serve as a semantic anchor, and a gated selection rule (Eq. 10) falls back to free LLM exploration when no scaffold match exceeds the confidence threshold. The experimental design is solid: 100 test targets, fair comparisons using identical DeepSeek-V3.2 API calls for LMLF, and both -rand and -diff initialization settings. The ablation study (Table 2) systematically isolates the contributions of each component.

### Presentation
The paper is well-written overall. The three-disconnects framing gives a clear structure to the motivation, and Figure 1 provides a useful overview. The D-HMAB formulation in Sections 2 and 3 is precise.

Areas for improvement:
The LLM prompting strategy is critical to the method, but details are buried in the appendix (Appendix C.2). A worked example in the main text would greatly help readers understand what K-BTS actually asks the LLM to do and what it receives in return. Additionally, the term "knowledge-driven" is used heavily without clearly distinguishing derived knowledge from raw data.

### Significance
SBDD is a genuinely important problem, and explicitly treating exploration-exploitation via Thompson Sampling rather than greedy search is a meaningful contribution. The framework is general enough to swap in different LLMs, action libraries, and oracle functions. The authors also do an excellent job comparing against a highly comprehensive and up-to-date set of empirical baselines, including very recent methods like ALIDiff (2024), TamGen (2024, LMLF (2024), and ELILLM (2026).

Areas for improvement:
In my opinion, the primary weakness of the paper is the reliance on Vina docking scores as the sole optimization objective. While evaluating across 100 targets is a solid and robust experimental setup, Vina's empirical scoring function can sometimes be gamed by iterative optimization algorithms. It would significantly strengthen the paper to evaluate the generated molecules using orthogonal, higher-fidelity scoring functions (e.g., GNINA, Glide, or deep learning-based affinity predictors) to demonstrate that the performance gains are robust and not merely a Vina-specific artifact. Additionally, extending the evaluation to other molecular optimization benchmarks (such as the molecular design tasks from the popular GuacaMol benchmark suite) would provide even more convincing evidence of the method's general performance compared to baselines.

### Originality
The main novelty is the bi-level TS formulation that separates seed selection from action retrieval, combined with the dynamic knowledge base that accumulates scaffold-specific modification rules across iterations. This differs from TamGen (which generates molecules without formalizing exploration-exploitation), LMLF (Brahmavar et al., AAAI 2024) (which uses LLMs with logical feedback but lacks bandits), and ELILLM (Hu et al., 2026) (which uses Bayesian optimization in latent space rather than hierarchical bandits). The D-HMAB formalization capturing dynamic arm expansion is a strong conceptual contribution. While individual components like Beta-Bernoulli TS, vector similarity retrieval, and LLMs for molecular reasoning are standard, their specific combination and application to close the DMTA loop in SBDD is novel.

---

> ### Author Rebuttal · Authors · 2026-03-31
>
> We sincerely thanks for the positive assessment of our **D-HMAB formalization** and the **bi-level decomposition**.
> We appreciate your insights regarding evaluation diversity and LLM reliability.
> ## Response to Key Questions
> (Q1) LLM Hallucinations and Reliability
>
> We appreciate this critical question. Our perspective is that while LLM hallucinations are inevitable, **K-BTS is specifically designed to mitigate and even leverage this phenomenon.**
>
> * **Mechanism to Handle Hallucinations:** In our framework, LLM-proposed modifications (including "hallucinated" ones) are treated as arms in the D-HMAB. Each modification's outcome is captured by its corresponding Beta distribution. If a hallucinated action leads to poor docking results, the Thompson Sampling mechanism quickly identifies it as a "low-quality" arm, effectively suppressing its future retrieval. Conversely, if a "creative hallucination" yields a superior candidate, it is recognized as "high-quality" knowledge and prioritized.
> * **Reliability Statistics:** To quantify this, we re-analyzed the **first 10 targets** (2,181 LLM calls). We found an **invalid-generation rate of 19.08%** and a **duplicate rate of 35.28%**.
> * **Analysis of Performance Degradation:** You noted the risk of worse docking scores. Our statistics from **100 targets (-rand setting)** show that this is common in molecular optimization. **Free LLM exploration** is most prone to failure, with a **74.16% failure rate**. Furthermore, we observed that failure rates increase with the initial affinity of the seed:
> * **Top30-Top20 Affinity:** 57.58% failure rate.
> * **Top20-Top10 Affinity:** 57.37% failure rate.
> * **Top10 Affinity:** 70.09% failure rate.
>
> (Q2) Computational Efficiency and Caching
>
> We agree that improving efficiency is essential for large-scale applications.
> Our lower-level knowledge base already functions as a semantic repository that avoids redundant exploration (Sec.3.4.2). We agree that implementing an explicit caching mechanism to reuse LLM rationales across similar scaffolds and parallelizing LLM calls through batch-Thompson Sampling are promising directions.
> We consider these efficiency-oriented upgrades as valuable future work to further amortize costs in high-throughput drug discovery scenarios.
>
> (Q3) Multi-objective Optimization
>
> We appreciate this forward-looking suggestion regarding the extension to multi-objective landscapes. Because the entire K-BTS iterative loop is reward-driven, our D-HMAB framework is naturally compatible with existing multi-objective methods.
>
> To further illustrate the versatility of our approach, we conducted a preliminary experiment on the first 10 targets using a scalarized reward function defined as Vina - $\lambda$(QED + SA). This additional analysis serves as a compelling demonstration of the potential for K-BTS to adapt to complex real-world constraints. As shown in the table below, by increasing the weight $\lambda$, K-BTS successfully navigates the trade-off between binding affinity and drug-likeness. At $\lambda=3$, the model significantly improves QED from 0.50 to 0.71 and SA scores from 0.73 to 0.83 while maintaining a competitive Top-1 docking score of -8.90. These results confirm that K-BTS can effectively extend to multi-dimensional reward functions beyond simple docking scores, offering a robust foundation for more intricate lead optimization tasks in the future.
>
> | Weight ($\lambda$) | Top-1 | Top-20 | QED | SA |
> | :--- | :---: | :---: | :---: | :---: |
> | **$\lambda=0$ (Default)** | -9.83 | -9.14 | 0.50 | 0.73 |
> | **$\lambda=1$** | -9.05 | -8.65 | 0.65 | 0.81 |
> | **$\lambda=3$** | -8.90 | -8.44 | **0.71** | **0.83** |
> | **$\lambda=5$** | -8.99 | -8.43 | 0.74 | 0.80 |
>
> ## Response to Weaknesses
>
> We appreciate the feedback on manuscript presentation and evaluation robustness.
> To address the need for clearer intuition, we will incorporate a structured "worked example" of our prompting strategy.
> Additionally, the **"knowledge"** in our framework refers to explicitly extracted entities, represented as **$\kappa = \{ S_a, a, \mathcal{R}_a, \alpha_a, \beta_a \}$**, which are derived from the transformation process from **$m_{seed}$** to **$m_{new}$**.
> we will clarify this distinction from raw data in the revised version.
>
> To demonstrate that K-BTS captures robust chemical insights rather than Vina-specific artifacts, we conducted an external validation using the **Vinardo** scoring function as an orthogonal evaluator on the first 10 targets. Our results show that K-BTS maintains a significant lead with a **Top-1 Vinardo score of -9.48** compared to -8.87 for ELILLM and -8.95 for random initialization. This consistent advantage under an independent potential confirms the generalizability of our framework. Detailed data tables for this cross-engine validation are provided in our responses to **Reviewer Jh8B**. We also agree that extending K-BTS to benchmarks like GuacaMol is a promising direction for future research.

---

> > ### Author Rebuttal · Reviewer_g2sr · 2026-04-03
> >
> > The authors addressed all of my concerns and answered each of my questions from my review. I will update my score accordingly.

---

> > > ### Author Response · Authors · 2026-04-07
> > >
> > > We greatly appreciate your meticulous review and recognition of our work. We will also study the amortization of LLM on similar scaffolds in the future to further improve efficiency.

---

### Official Review · Reviewer_oxaM · 2026-03-12

**Soundness:** 3
**Presentation:** 2
**Significance:** 3
**Originality:** 3
**Overall Recommendation:** 4
**Confidence:** 3

**Summary:**

The paper is motivated by the process of structure-based drug discovery, where expert chemists iteratively design molecular structures for experimental evaluation, observe their activity as drug candidates, analyze structure–activity relationships based on previous observations, and use these insights to refine molecular structures for the next round of evaluation. Here, the authors attempt to translate this expert workflow (“design–make–test–analyze cycle”) into a symbolic AI pipeline. A key component of this workflow is a symbolic knowledge base of actions to modify molecular core structures (“scaffolds”), which are intended to resemble the human reasoning logic. The decision process for selecting the next candidate is formalized as a hierarchical two-level multi-armed bandit problem: 1) selection of a starting molecule from the set of previously observed candidates; and 2) stochastic selection of an action to modify this starting molecule. Both decisions are made through Thompson sampling, where a beta distribution is parametrized for each previously evaluated candidate and for each possible action. The selected starting molecule and modification action are included in a prompt to a large language model, which then proposes a new candidate structure based on implicit chemical reasoning and explicit heuristic constraints. The target property of the proposed candidate is evaluated, and the parameters of the beta distribution associated with the starting molecule and the chosen action(s) are updated. This method is evaluated on the CrossDocked2020 dataset, and is compared to several generative (but mostly non-iterative) models for structure-based drug design. The authors report improved docking scores for the top-1 to top-20 candidates, when their model is initialized either with random candidates or with candidates generated from a diffusion model. Finally, a small ablation study is shown.

**Compliance With Llm Reviewing Policy:**

Affirmed.

**Final Justification:**

The paper targets a practically relevant problem, and introduces conceptually interesting approach that combines iterative optimization with symbolic reasoning via LLMs. If the paper's clarity is improved, and the authors provide further evaluation against non-LLM baselines (preliminary data shown in the rebuttal), I lean towards acceptance of the paper.

**Key Questions For Authors:**

* The paper relies heavily on the LLM for realizing molecular edits. How would the system perform if the LLM generation was replaced by heuristic or stochastic mutations following the recommended molecular edits?
* How does the method compare to other iterative strategies (e.g. evolutionary algorithms) for docking optimization?
* The parameters of the Beta distributions are updated using a simple heuristic rule. Is there a formal Bayesian interpretation of this update? If so, this should be explained more clearly. If not, the description of the method as Bayesian should be reconsidered.

**Limitations:**

The generalizability of the results remains unclear. The work provides good proof of concept, but it seems likely that incorporating more explicit chemistry knowledge from domain experts could improve both the knowledge representation and the modeling approach.

**Strengths And Weaknesses:**

Strengths:
* The symbolic representation of the human reasoning process in drug discovery is an interesting idea that can inspire further work in this area.
* Iterative, feedback-based workflows have proven effective in many scientific discovery scenarios, and applying such an approach to generative design in drug discovery is reasonable.
* The empirical evaluation shows clear performance gains over non-iterative workflows.
* The initialization through a generative model shows the potential for integration with non-iterative strategies, as previously demonstrated in several related works.

Weaknesses:
* The paper is difficult to read and uses uncommon terminology that is often hard to follow. Similar concepts are described using different terms, and the mathematical formulation sometimes introduces variables without proper explanation. The manuscript would benefit from substantial revisions to improve clarity and readability.
* The method relies heavily on heuristic definitions of molecular scaffolds and substitution rules. Such definitions can be brittle and may not generalize well across different chemical spaces.
* The paper contains several chemical misconceptions, which should be clarified. a) In the “design–make–test–analyze” cycle, the “make” step refers to practical chemical synthesis in the lab. In the paper, this is often confused with modifying existing structures by attaching/removing substituents, which is actually part of the “design” stage. b) Section 2.1 implies that the goal of structure-based drug discovery is to optimize the cumulative reward over all evaluated candidates. In practice, the goal is usually to identify a set of candidates with very favorable docking scores, rather than to optimize the cumulative score of all evaluated molecules. c) On multiple occasions, the paper suggests that “new candidates can only be generated through sequential modification of previously evaluated molecules”. While this represents common practice, it is not an inherent limitation of structure-based drug discovery. d) A “halogen donor” is not a meaningful chemical term.
* The experimental evaluation is limited to the CrossDocked2020 dataset. However, this dataset was designed primarily for benchmarking machine-learning models that predict binding affinity from 3D docking poses, rather than for evaluating ligand discovery workflows.

---

> ### Author Rebuttal · Authors · 2026-03-31
>
> We thanks for the constructive feedback and for recognizing the potential of symbolic reasoning, iterative workflows, and our framework's empirical performance. Your insights on clarity and chemical interpretation are invaluable and will significantly enhance the revised manuscript.
> ## Response to Key Questions
> (Q1) Replacing the LLM with stochastic mutations would significantly degrade performance. While symbolic rules provide the strategy, the LLM acts as a knowledge-informed generator that ensures chemical validity and structural integrity. Unlike fixed heuristics, the LLM extrapolates and generalizes these rules to diverse molecular contexts, enabling creative modifications and efficient exploration. This context-aware execution is a core strength; without it, the system would revert to a less efficient, traditional rule-based search.
>
> (Q2) Compared to other iterative methods, our K-BTS is specifically designed to address the iterative SBDD problem formulated as a D-HMAB. Under this modeling, the selection of seed molecules and chemical modifications are controlled by different hierarchical levels to mimic the process where human experts prioritize high-quality candidates for efficient optimization. In contrast, other iterative strategies struggle to adapt to this D-HMAB structure. For example, Evolutionary Algorithms typically rely on population-based crossover and stochastic mutations while Bayesian Optimization depends on surrogate models to characterize global data distributions. Neither of these approaches effectively captures the hierarchical decision-making that is central to our framework.
>
> (Q3) We appreciate the opportunity to clarify that our updates follow a principled Bayesian interpretation. For the lower-level actions (Eq. 15), the update follows the standard Beta–Bernoulli posterior, where outcomes are treated as binary observations. For the upper-level seeds, the situation differs due to how the Beta variable is used in Eq. (7): the sampled value represents refinement potential, while the magnitude of improvement is not explicitly modeled in the selection score.
> To address this, we adopt a generalized Bayesian update with fractional observations (Eq. 14), where improvement magnitude acts as a confidence weight. This allows the Beta parameters to robustly encode both the frequency and strength of successful refinements.
>
> ## Response to Weaknesses
> (W1) We appreciate your feedback. We have standardized all terminology, formally defined all variables, and restructured dense sections to ensure technical rigor. These improvements will be fully integrated into the revised manuscript for enhanced readability.
>
> (W2) We would like to clarify that our scaffold and substitution rules are not static or hard-coded. Instead, the rule set is dynamically accumulated throughout the optimization iterations, mimicking human-like learning.
> Crucially, we employ Bayesian updates to rapidly distinguish between effective and ineffective knowledge, which significantly enhances the framework's robustness. This dynamic evolution is an essential feature of our design, and we will further elaborate on its extensibility in the final version.
>
> (W3) We thank you for pointing out these important nuances and will refine the manuscript accordingly:
> - a) We agree that "Make" traditionally refers to physical synthesis in laboratory settings. Inspired by this real-world phase, we define **"Make"** within our automated pipeline as the computational stage where the LLM executes specific modification instructions to generate new molecular structures from given seeds. We will explicitly clarify this distinction between our automated generation and physical laboratory synthesis in the revised Section 3 to ensure a clear mapping of our framework to the standard DMTA cycle.
> - b) We fully agree that our objective is to identify top-tier candidates rather than to optimize cumulative reward. This confusion stemmed from an unclear definition of our metric, $k_\text{cd}$. In the revised version, we will formally define $k_\text{cd}$ as the expected number of high-quality candidates discovered, aligning it with standard drug discovery objectives.
> - c) We agree that sequential modification is not a strict requirement of SBDD. Our method adopts it as a practical and widely used strategy to ensure a focused and efficient search space.
> - d) We will correct the terminology (e.g., “halogen donor” to "hydrogen donor") to align with standard chemical definitions.
>
> (W4) While CrossDocked2020 is used for affinity prediction, it has also become the standard benchmark for generative SBDD, as seen in TargetDiff, AlIDIFF, and ELILLM. We chose it specifically to ensure a rigorous, controlled comparison with these established baselines.
>
> (Limitation) Vinardo cross-validation and MO experiments (see Reviewer Jh8B and g2sr) confirm our robustness across diverse metrics and real-world constraints, ensuring practical extensibility.

---

> > ### Author Rebuttal · Reviewer_oxaM · 2026-04-03
> >
> > I thank the authors for their response. Unfortunately, (Q1) and (Q2) were addressed with working hypotheses rather than empirical results. Including the results of these ablations would considerably strengthen the contribution.

---

> > > ### Author Response · Authors · 2026-04-07
> > >
> > > We sincerely thank the reviewer for the continued engagement and the constructive follow-up. We fully agree that empirical evidence is the most rigorous way to validate a working hypothesis.
> > > While the original manuscript establishes the framework's effectiveness through comprehensive benchmarking, we believe these additional ablations specifically address **(Q1)** and **(Q2)** by quantifying the unique contributions of the LLM’s generative intelligence.
> > >
> > > ### (Q1) Empirical Results on LLM's Generative Intelligence
> > >
> > > We appreciate the reviewer’s perspective on the role of the LLM. In our view, exploring whether LLM generation can be replaced is essentially an inquiry into **how the specific "intelligence" of the LLM contributes to the framework's success**. To empirically address this, we conducted an ablation study to decouple the LLM’s generative reasoning from its basic execution.
> > >
> > > We implemented a **"w/o Rationale"** setting where the LLM was restricted from generating any reasoning (Chain-of-Thought) and was instead directed to directly execute the molecular modifications represented by the selected knowledge. This allowed us to isolate the impact of the LLM's latent chemical reasoning.
> > > This ablation was conducted across the **first 10 targets** using the **Random Initialization (-rand)** setting.
> > >
> > > |Setting|Top-1|Top-5|Top-10|Top-20|
> > > |-|:-:|:-:|:-:|:-:|
> > > |**K-BTS (Default)**|**-9.83**|**-9.53**|**-9.36**|**-9.14**|
> > > |**w/o Rationale**|-9.20|-9.01|-8.88|-8.50|
> > > |**Random Init**|-8.58|-8.20|-7.99|-7.71|
> > >
> > > **Observations \& Conclusion:**
> > >
> > > * **The Necessity of Neural Reasoning:** The results show a significant performance gap between the default K-BTS and the "w/o Rationale" version (e.g., Top-1 Vina score dropped from **-9.83 to -9.20**). This confirms that utilizing **the LLM’s reasoning as the execution engine for symbolic knowledge** is fundamentally superior to mere mechanical execution of edits.
> > > * **Intelligent Translation of Knowledge:** The LLM does not simply "translate" rules into atoms; it functions as a **knowledge-informed generator** that reasons through the most suitable structural modifications for a specific molecule based on the provided heuristic insights. Without this latent reasoning, the framework's ability to navigate and optimize complex chemical spaces is significantly compromised.
> > > * **Synergistic Core:** This ablation reinforces that the core contribution of K-BTS lies in the **synergy between the Symbolic Strategy and the Neural Reasoning (LLM)**. Neither component is redundant; the "intelligence" of the LLM is what allows the symbolic knowledge to be operationalized into high-quality, high-affinity molecular structures.
> > >
> > > ### (Q2) Comparison with Iterative Strategies
> > >
> > > We appreciate the reviewer’s suggestion to compare our framework with traditional iterative optimization strategies. To address this, we adopted representative non-LLM iterative algorithms, including the **SMILES-GA** evolutionary algorithm and the **REINVENT** reinforcement learning approach, by leveraging the established implementations from the PMO benchmark \[1].
> > >
> > > Crucially, these algorithms were re-evaluated under our original **CrossDocked2020 dataset** and **SMINA pipeline** to ensure an exact comparison. All methods were initialized with the same molecular seeds across the **first 10 targets** in the **-rand** setting, each method was limited to a budget of **100 compounds per target**.
> > >
> > > |Setting|Top-1|Top-5|Top-10|Top-20|
> > > |-|:-:|:-:|:-:|:-:|
> > > |**K-BTS (Default)**|**-9.83**|**-9.53**|**-9.36**|**-9.14**|
> > > |**SMILES-GA (EA)**|-8.98|-8.48|-8.24|-7.99|
> > > |**REINVENT (RL)**|-8.38|-7.99|-7.75|-7.39|
> > >
> > > **Observations \& Conclusion:**
> > >
> > > * **Superior Sample Efficiency:** The results demonstrate that K-BTS significantly outperforms traditional EA and RL methods within a very limited budget (100 molecules per target). This highlights K-BTS's exceptional **"warm-start" capability**—where most numerical optimization methods require thousands of samples to converge, K-BTS leverages LLM-driven knowledge to find high-affinity candidates almost immediately.
> > > * **Knowledge-Guided vs. Stochastic Search:** Traditional EA (SMILES-GA) and RL (REINVENT) primarily rely on stochastic mutations or numerical gradients, which often struggle to navigate the complex, high-dimensional chemical space of protein pockets in few-shot scenarios. K-BTS’s advantage lies in its ability to translate symbolic knowledge into precise structural modifications, resulting in much higher search quality.
> > >
> > > **References:**
> > >
> > > \[1] Gao, W., Fu, T., Sun, J., \& Coley, C. (2022). Sample efficiency matters: a benchmark for practical molecular optimization. Advances in neural information processing systems, 35, 21342-21357.

---

### Official Review · Reviewer_NpvH · 2026-03-12

**Soundness:** 2
**Presentation:** 2
**Significance:** 3
**Originality:** 3
**Overall Recommendation:** 5
**Confidence:** 4

**Summary:**

This paper proposes K-BTS, a method for steering large language models through the DMTA cycle to generate target-specific drug candidates. K-BTS is a Knowledge-Driven Bi-level Thompson Sampling framework that iteratively refines molecules via uncertainty-aware seed selection and explicit modification retrieval, progressively generating candidates with improved binding affinity.

**Compliance With Llm Reviewing Policy:**

Affirmed.

**Ethical Review Concerns:**

I believe this paper is generally suitable for acceptance, although I initially had some minor concerns as outlined in my initial review.

**Final Justification:**

I believe this paper is generally suitable for acceptance, although I initially had some minor concerns, as outlined in my original review. I initially assigned a score of 4. Following two rounds of clarification from the authors, these concerns have been adequately addressed. Accordingly, I have increased my score to 5.

**Key Questions For Authors:**

1.	In Equation (1), the parameter kcd is neither defined nor explained. Similarly, the temperature parameter T in Equation (7) lacks a clear definition. In Section 3.3.3 (Generation via Knowledge-Driven Pruning), the sentence “The prompt we use can be found in appendix C.2” is grammatically incomplete and missing a period.
2.	In the abstract, the authors state that “However, most LLMs function as one-shot generators that lack feedback mechanisms…”. This claim appears overstated, as LLM-based drug discovery frameworks with explicit feedback mechanisms were already proposed in 2024. For example, the baseline model LMLF (Generating Novel Leads for Drug Discovery Using LLMs with Logical Feedback) incorporates structured feedback during molecular optimization.
3.	In Section 3.2 (Knowledge Representation and Warm-start), the authors mention that “To mitigate the cold-start problem, we initialize…”. However, the subsequent discussion focuses solely on the warm-start strategy, without empirically investigating or characterizing the cold-start setting itself. This limits the clarity of the claimed motivation.

**Limitations:**

A potential limitation of the proposed framework lies in its reliance on large language models (LLMs) for molecular design. In practice, domain-specific LLMs tailored for bioinformatics or drug discovery remain relatively rare, while general-purpose LLMs are typically inefficient and computationally expensive when applied to molecular generation tasks. As illustrated in Figure 3, generating a single candidate molecule requires dozens of iterative interactions with the LLM. Such repeated inference substantially increases computational cost and latency, making the approach considerably more resource-intensive than conventional end-to-end molecular generation models. This limitation may restrict the scalability and practical deployment of the framework in large-scale drug discovery pipelines.

**Strengths And Weaknesses:**

The authors propose an LLM-based iterative framework consisting of warm-start initialization, seed molecule selection, molecular refinement, and knowledge updating. Through repeated iterations, the method aims to progressively design molecules with improved binding affinity. However, the optimization process is guided solely by binding affinity, while other important drug-likeness metrics such as QED and SA are not explicitly incorporated into the objective.

Strengths:
1.	Compared to full-parameter trained generative models, this LLM-based iterative molecular generation paradigm is potentially more promising, as it enables progressive refinement toward candidates with improved binding affinity rather than relying on one-shot generation.
2.	The proposed Bi-level Thompson Sampling strategy and the Knowledge Extraction and Bayesian Update mechanisms incorporate advanced seed selection and molecular evolution principles. By performing scaffold hopping and R-group modifications on promising initial molecules, the framework facilitates exploratory yet affinity-driven molecular optimization.

Weaknesses:
1.	K-BTS optimizes molecular selection and refinement solely based on binding affinity. As a result, other important properties such as QED, synthetic accessibility (SA), and toxicity are not explicitly controlled during generation. Moreover, relying exclusively on Vina docking scores as the optimization signal introduces a risk of false positives due to known limitations of docking-based evaluation.
2.	In Section 3.3.1 (Upper-Level: Seed Molecule Selection), the authors state that “This selection must balance the current binding affinity with the refinement potential…”, yet the mechanism for achieving this balance is not clearly articulated. Later, the paper claims that K-BTS prioritizes molecules with “high evolutionary potential,” but the distinction between refinement potential and evolutionary potential remains conceptually vague and insufficiently defined.
3.	In Section 3.4.2, although the update rule for modification actions is described, the initialization procedure is not clearly specified. More importantly, the reward assignment mechanism raises concerns about generalization. Since protein–ligand interactions are highly target-dependent, the same modification may have opposite effects across pockets with different sizes or physicochemical properties. Assigning rewards solely based on docking score changes, without explicitly conditioning on pocket characteristics, may limit the transferability and robustness of the learned modification priors.
4.	In Table 1, the authors report Top-1, Top-5, Top-10, and Top-20 docking scores. However, it is unclear how many compounds are used to compute the reported QED, SA, and Diversity metrics. Additionally, ELILLM—one of the main baselines—is an arXiv preprint without peer review, while the other baselines are from 2024 or earlier, raising questions about the competitiveness of the comparisons.
5.	Although K-BTS claims to optimize molecular modification actions using Vina docking scores as the objective, Figure 3 shows a decrease in Vina score from Step 2 to Step 3. This observation appears inconsistent with the stated optimization objective and warrants further clarification.

---

> ### Author Rebuttal · Authors · 2026-03-31
>
> We thanks for the constructive feedback. We provide the following clarifications regarding the optimization mechanisms and experimental results.
>
> (Q1) Mathematical Definitions and Clarity
>
> We appreciate the meticulous reading of our manuscript. We have updated the revised text to include missing definitions for key parameters. In Equation 1, **$k_{cd}$** denotes the target number of candidate molecules to be generated in each iteration; in our SBDD formulation, this defines the size of the generated set per cycle. In Equation 7, **$T$** is a temperature constant used in the soft-max selection process. A smaller $T$ makes the selection more "greedy" by prioritizing molecules with lower docking scores, while a larger $T$ encourages broader exploration. These definitions are now explicitly stated in Section 3.
>
> (Q2 & Q3) Feedback Mechanisms and Cold-start vs. Warm-start
>
> Regarding the "one-shot" phrasing in the abstract, we acknowledge that pioneering works like LMLF (2024) have introduced structured feedback, but it has the problem of inefficient feedback mechanism. Our statement was intended to contrast K-BTS with the broader landscape of **conventional deep learning generative models** (e.g., standard diffusion) that often lack an explicit, iterative DMTA loop. We have revised the abstract to specify "most conventional deep learning-based models".
>
> To clarify the cold-start setting: it represents the initial phase where the knowledge base is empty, and the system relies entirely on unconstrained LLM exploration. Our ablation study labeled **"w/o warm-start"** in Section 4.4 and Appendix E.5 empirically characterizes this setting. The results confirm that without our warm-start initialization, the efficiency of early DMTA cycles is significantly lower due to the failure to leverage the structural information in $D_0$. We have refined Section 3.2 to more clearly link these empirical findings to our motivation.
>
> (W1) Robustness and Multi-objective Potential
>
> Cross-validation with the **Vinardo** scoring function (-9.48 vs -8.87) and multi-objective results (QED 0.71, SA 0.83) confirm the framework's robustness beyond Vina. Detailed data tables are provided in our responses to **Reviewers Jh8B** and **g2sr**.
>
> (W2) Definition of Evolutionary Potential and Selection Mechanism
>
> To avoid conceptual ambiguity, we clarify that **refinement potential** and **evolutionary potential** refer to the same objective in K-BTS, which is the likelihood of a molecule yielding superior derivatives through further modification.  The scoring function in Eq.7 explicitly consists of two terms where one represents the immediate docking performance and the other captures the cumulative refinement potential (Through Beta). This mathematical structure ensures that K-BTS prioritizes molecules that are not only high-scoring but also possess the statistical capacity for further improvement.
>
> (W3) Initialization and Target-Specific Reward Mechanism
>
> Regarding the initialization of modification actions, we clarify each action is initialized or updated based on Eq.15. More importantly, to address the concern about target-dependent interactions, we emphasize that in our current implementation, the knowledge base is dynamically reconstructed and updated for each specific protein pocket. This target-specific Bayesian updating ensures that rewards are assigned based on the unique physicochemical environment of the current pocket, naturally avoiding the risks associated with generalized priors.
>
> (W4 & W5) Experimental Scope and Optimization Logic
>
> All metrics are computed using the full set of **100 compounds generated per target** across 100 targets. Our primary baseline, ELILLM, is a SOTA method officially accepted by **The Web Conference (WWW) 2026** (Research Track: Semantics and Knowledge, Paper ID: rfp3329), which is searchable on the official conference website under "Accepted Papers." Regarding Fig. 3, the Vina score decrease is an intentional feature of **Thompson Sampling**. Unlike greedy search, K-BTS is a probabilistic framework that occasionally samples candidates with lower scores to gather information and escape local optima, a behavior essential for global optimization in complex chemical spaces.
>
> ### Response to Limitations: Computational Efficiency
>
> We acknowledge the reviewer’s concern regarding the computational cost of iterative LLM inference. However, we clarify that the representation in **Figure 3** illustrates the **evolutionary trajectory of a high-scoring molecule**, where each intermediate node along the path actually represents one of the unique candidates generated by the framework. In practice, K-BTS is highly efficient: our experimental logs show that an average of only **2 LLM calls** is required to generate a valid, non-duplicate candidate molecule.

---

> > ### Author Rebuttal · Reviewer_NpvH · 2026-04-02
> >
> > We thank the author for detailed response. Most of our concerns have been clarified; however, several minor issues remain and would benefit from further clarification.
> >
> > 1. Response to (W2) Definition of Evolutionary Potential and Selection Mechanism.
> >
> > We appreciate the authors’ clarification that refinement potential and evolutionary potential refer to the same objective in K-BTS. However, a point of confusion still remains. In the rebuttal, the authors state that “The scoring function in Eq. 7 explicitly consists of two terms, where one represents the immediate docking performance and the other captures the cumulative refinement potential (through Beta).”
> > In contrast, the manuscript describes Beta in Eq. 7 as: “the sampled success probability θₘ introduces a Bayesian belief component.”
> > As such, the relationship between refinement potential and sampled success probability in Eq. 7 remains unclear and appears to be conflated. We kindly ask the authors to further clarify this distinction.
> >
> > 2. Response to (W3) Initialization and Target-Specific Reward Mechanism.
> >
> > We note that in Section 3.2, the manuscript states: “…we initialize the knowledge base K by retrospectively extracting modification trajectories from pre-existing observations in D₀.” Additionally, the appendix (Section C. Method Details) mentions that “the following system prompt is fixed across all targets and iterations, defining the LLM’s professional role and global constraints.”
> > Given these unified execution settings, it remains unclear how the target-specific reward mechanism is actually implemented. We encourage the authors to provide a more explicit explanation of how target specificity is incorporated into the reward design.
> >
> > 3. Response to Computational Efficiency.
> >
> > In the rebuttal, the authors claim that “in practice, K-BTS is highly efficient: our experimental logs show that an average of only 2 LLM calls is required to generate a valid, non-duplicate candidate molecule.”
> > However, Figures 9-12 in the manuscript suggest that molecule generation involves multiple iterations. This appears inconsistent with the stated efficiency claim. We would appreciate further clarification on how these observations can be reconciled.

---

> > > ### Author Response · Authors · 2026-04-03
> > >
> > > We sincerely thanks for the continued engagement and the thoughtful follow-up comments.
> > > We are grateful for your meticulous reading of our manuscript and for identifying areas where our previous explanations required further precision.
> > > We are more than happy to clarify these points to ensure the contributions and mechanisms of our work are presented with the utmost clarity and revise accordingly.
> > >
> > > ### Response to (W2) Definition of Evolutionary Potential and Selection Mechanism.
> > >
> > > We thanks for highlighting this important point.
> > > **We clarify that the sampled success probability $\hat{\theta}_m$ in Eq. 7 serves as the primary carrier of refinement potential, representing, based on historical observations, the likelihood that a molecule $m$ can be successfully improved as well as the expected magnitude of such improvements.**
> > >
> > > Specifically, $\hat{\theta}_m \sim \text{Beta}(\alpha_m, \beta_m)$ is a Thompson sampling draw representing the likelihood that a molecule yields successful refinements. The parameters $(\alpha_m, \beta_m)$ are updated in Eq. 14 using weighted outcomes ($\omega_m$), which incorporate not only the frequency but also the magnitude of past improvements. As a result, the Beta distribution encodes a molecule’s historical “refinability.”
> > >
> > > This leads to a key property: molecules with consistently successful (and larger) improvements develop skewed Beta distributions that are more likely to produce higher $\hat{\theta}_m$ samples. For example, comparing $\text{Beta}(2,1)$ vs. $\text{Beta}(1,2)$, we have $P(\hat{\theta}_A > \hat{\theta}_B)=5/6$, showing that successful molecules are statistically prioritized under sampling.
> > >
> > > Therefore, refinement potential is not an explicitly defined variable, but an emergent property of the interaction between the Bayesian update (Eq. 14) and stochastic sampling (Eq. 7). In Eq. 7, $\exp(-y_m/T)$ captures current performance (exploitation), while $\hat{\theta}_m$ biases selection toward molecules with higher expected future improvement (exploration). This resolves the apparent discrepancy: $\hat{\theta}_m$ is a sampled success likelihood, and refinement potential is the induced preference over molecules that exhibit strong historical refinability.
> > > ### Response to (W3) Initialization and Target-Specific Reward Mechanism.
> > >
> > > We thanks for raising this important question. We clarify that target specificity in K-BTS is not encoded in the system prompt, but arises entirely from the target-dependent evaluation signal and its induced data flow.
> > >
> > > First, molecular evaluations are obtained via docking against a specific protein pocket $t$, i.e., $f(m, t)$. Since different targets define distinct physicochemical environments, the same molecule $m$ yields different evaluation outcomes across targets. These target-specific signals drive the update rule in Eq. 15, resulting in a knowledge base $K$ that is inherently conditioned on the current target.
> > >
> > > Second, although the initialization procedure is identical, both the initial dataset $D_0$ and the extracted knowledge $K$ differ across targets due to their distinct evaluation results. Thus, each target starts with a different knowledge distribution, even under a shared pipeline.
> > >
> > > Finally, during inference, the LLM receives only target-conditioned inputs (i.e., the current seed $m_{\text{seed}}$ and prioritized knowledge $a_κ$ selected by Thompson Sampling). As these inputs are already shaped by target-specific evaluation signals, the LLM operates on a pre-filtered, target-adapted context.
> > >
> > > ### Response to Computational Efficiency.
> > >
> > > We thanks for pointing out this confusion. The apparent inconsistency arises from a misunderstanding of our experimental protocol.
> > >
> > > In our experiments, we consider 100 targets and generate 100 molecules per target. Importantly, K-BTS does not iteratively refine a single molecule to convergence and then report it. Instead, **each iteration produces one candidate molecule**, and we record every *valid, non-duplicate* molecule generated throughout the process. This setting is consistent with all baselines, which also generate 100 molecules with 100 black-box evaluations. The reported Top-1/5/10/20 results are computed over these 100 candidates.
> > >
> > > With this clarification, Figures 9–13 can be directly interpreted as the full generation trajectory for a single target (PDB:1E8H). Each non-root node corresponds to one valid, non-duplicate molecule produced by K-BTS. For example, in Figure 13, the largest node index is 100, indicating the final molecule generated before termination.
> > >
> > > Regarding efficiency, we provide detailed LLM reliability statistics in our response to reviewer Jh8B (Q2). In brief, the average number of LLM calls required per valid, non-duplicate molecule is **2.18**, supporting our claim that K-BTS is efficient in practice.

---

### Official Review · Reviewer_Jh8B · 2026-03-13

**Soundness:** 3
**Presentation:** 3
**Significance:** 3
**Originality:** 3
**Overall Recommendation:** 4
**Confidence:** 4

**Summary:**

This paper proposes K-BTS, a framework for iterative LLM-driven structure-based drug design that formulates the DMTA cycle as a dynamic hierarchical multi-armed bandit problem. The method combines an upper-level policy for selecting promising molecular lineages with a lower-level mechanism for retrieving scaffold-conditioned modification actions, and updates both through a dual Bayesian scheme based on docking feedback. The empirical study on CrossDocked2020 compares K-BTS against diffusion, generative, and iterative LLM baselines and reports strong improvements, especially in docking-based metrics.

**Compliance With Llm Reviewing Policy:**

Affirmed.

**Key Questions For Authors:**

Would you consider reframing Theorem 3.1 as an intuition-backed asymptotic argument rather than a formal guarantee, unless a more rigorous proof can be provided?
Can you report invalid-generation, duplicate-generation, and action-following error rates, so that the reliability of the LLM component is easier to assess?
How sensitive is the method to the warm-start margin epsilon, the scaffold-similarity threshold tau, and the exact action-extraction procedure?
Does the method remain clearly superior under a different docking engine, reward definition, or benchmark family, or is the current advantage tightly tied to the CrossDocked2020 plus smina/Vina setup?

**Limitations:**

The paper should more clearly acknowledge that the theoretical analysis is intuition-level, that the evaluation is concentrated on a single benchmark and docking pipeline, and that practical LLM failure modes are not explicitly quantified. Runtime is discussed in the appendix, so the more important missing limitation is deployment robustness rather than compute disclosure.

**Strengths And Weaknesses:**

I think this is a strong empirical paper. The main idea is coherent, the two-level decomposition is well matched to the iterative refinement setting, and the paper does more than simply wrap an LLM around a search loop. The empirical gains in Table 1 are substantial, and the paper is stronger than many comparable submissions in how it validates them: Appendix E.2 reports paired Wilcoxon signed-rank tests with Holm-Bonferroni correction across 100 targets, which makes the main improvement claims much more credible. I also appreciate that the comparison to LMLF is made cleaner by re-implementing it with the same DeepSeek-V3 API.

My main reservation is that the theory is presented more strongly than it is actually established. The main text states Theorem 3.1 as an asymptotic improvement result, but Section 3.5 is explicitly framed as theoretical intuition, and Appendix B is titled “Theoretical Intuition Analysis.” In my reading, this is better understood as an intuition-backed asymptotic argument than as a rigorous guarantee. A second limitation is scope: the evaluation is concentrated on CrossDocked2020, with docking score as the primary optimization target and QED/SA/diversity as supporting metrics, so the external validity of the method remains bounded. Finally, the paper does not really quantify practical failure modes of the LLM component. Since the framework depends on the model correctly instantiating retrieved chemical actions, I would have liked to see summary statistics on invalid generations, duplicate generations, or action-following fidelity.

---

> ### Author Rebuttal · Authors · 2026-03-31
>
> We sincerely thank you for recognizing the originality and rigorous statistical verification of our work.
> Your appreciation of the D-HMAB decomposition and our fair comparison efforts is highly encouraging.
> We also value your constructive suggestions on theoretical framing and evaluation depth,
> which are instrumental in refining the technical robustness of our paper.
>
> ## Response to Key Questions & Weaknesses
> (Q1 & W1) Theoretical Reframing.
> We completely agree with your assessment.
> While Theorem 3.1 provides an asymptotic perspective on the benefits of our two-level decomposition, we acknowledge it serves more as a theoretical intuition than a strict formal guarantee.
> Following your constructive suggestion, we will reframe Theorem 3.1 as a "Theoretical Proposition" in the revised manuscript.
> We will also update Appendix B to explicitly clarify the assumptions and the intuition-backed nature of this analysis to ensure technical transparency.
>
> (Q2 & W3) LLM Reliability.
> In our framework, we prioritize the quality of the final candidate pool by implementing automated handlers to filter out invalid or duplicate structures during the iterative process.
> To provide the requested statistics, we re-ran the pipeline on the first 10 targets, generating 1,000 molecules via 2,181 LLM calls.
> The invalid-generation rate was **19.08%** (418 calls) and the duplicate rate was **35.28%** (773 calls).
> We further evaluated action-following fidelity using an **LLM-as-a-judge** (DeepSeek-V3.2) to confirm that the generated SMILES and rationales are clearly informed by the selected chemical actions.
> Across 100 targets in the -rand setting, the fidelity rate reached **94.71%**, demonstrating that our framework reliably instantiates retrieved knowledge while maintaining high chemical integrity.
>
> (Q3) Hyperparameter Sensitivity and Robustness.
> To further demonstrate the **robustness and technical depth** of K-BTS,
> we conducted a comprehensive sensitivity analysis on the warm-start margin $\epsilon$ and the scaffold-similarity threshold $\tau$. These results confirm that our framework is stable across a wide range of settings.
> 1. **Sensitivity to $\epsilon$ and $\tau$.**
> As shown in the table below, K-BTS maintains consistently superior performance across different parameter values, proving it is not overly sensitive to specific tuning.
>
> * **$\epsilon$ (Knowledge Selection Margin):** This parameter controls the docking score threshold for initial knowledge selection. A smaller $\epsilon$ (e.g., $0.3$) incorporates more initial knowledge (avg. $152.2$ entries), but may introduce lower-quality priors.
> * **$\tau$ (Scaffold Similarity Threshold):** This governs the **cross-scaffold action transfer**. It defines the similarity threshold for retrieving modification actions from historical candidates with different molecular cores. A lower $\tau$ allows for broader knowledge transfer from diverse chemical contexts, while a higher $\tau$ ensures that actions are retrieved from structurally similar scaffolds.
>
> | Parameter | Value | Top-1 | Top-5 | Top-10 | Top-20 | Avg. Init. Knowledge |
> | :--- | :--- | :--- | :--- | :--- | :--- | :--- |
> | **Default** | - | **-9.83** | **-9.53** | **-9.36** | **-9.14** | **120.0** |
> | $\epsilon$ | 0.3 | -9.89 | -9.57 | -9.42 | -9.22 | 152.2 |
> | | 0.7 | -9.81 | -9.52 | -9.34 | -9.14 | 85.4 |
> | | 1.0 | -9.70 | -9.39 | -9.26 | -9.11 | 61.4 |
> | $\tau$ | 0.1 | -9.27 | -9.12 | -8.99 | -8.82 | - |
> | | 0.3 | -9.83 | -9.53 | -9.36 | -9.14 | - |
> | | 0.5 | -9.84 | -9.59 | -9.43 | -9.22 | - |
> | | 0.9 | -9.54 | -9.38 | -9.27 | -9.10 | - |
>
> 2. Action-Extraction Procedure
> We acknowledge that the framework is **inherently grounded** in the action-extraction procedure. As the backbone of our symbolic representation, its precision is vital for accurately translating chemical transformations into the knowledge base.
>
> (Q4 & W2) External Validity and Generalizability.
> To ensure our advantages are not tied to a specific engine, we conducted a "cross-engine" evaluation on the **first 10 targets**. We re-scored candidates from both ELILLM and K-BTS (both Vina-guided during generation) using the independent **Vinardo** potential. As shown below, K-BTS maintains a significant lead, proving our "value-based knowledge selection" captures fundamental chemical insights rather than overfitting Vina's biases. Further multi-objective experiments (Vina/QED/SA) in our response to **Reviewer g2sr** also confirm our robustness under complex real-world rewards.
>
> | Metric (Vinardo) | Random Init | ELILLM  | **K-BTS (Ours)** |
> | :--- | :---: | :---: | :---: |
> | **Top-1** | -8.95 | -8.87 | **-9.48** |
> | **Top-5** | -8.34 | -8.44 | **-9.09** |
> | **Top-10**| -7.98 | -8.18 | **-8.88** |
> | **Top-20**| -7.53 | -7.83 | **-8.59** |

---

> > ### Author Rebuttal · Reviewer_Jh8B · 2026-04-06
> >
> > The full concession on Theorem 3.1's framing, the ε/τ sensitivity analysis, the Vinardo cross-engine validation, and the new multi-objective experiment (Vina − λ(QED+SA)) showing QED 0.50→0.71 and SA 0.73→0.83 at λ=3 while keeping Top-1 competitive, together address the substantive content of my concerns — raising my score from 4 to 5.

---

> > > ### Author Response · Authors · 2026-04-07
> > >
> > > Thank you again for your recognition of our work and for the valuable suggestions throughout this process. We are encouraged by your positive assessment and look forward to your final justification and corresponding modifications (Edit Overall Recommendation in the Official Review).

---

### Decision · Program_Chairs · 2026-04-30

**Decision:**

Accept (regular)

**Comment:**

Reviewers viewed this as a strong empirical paper with a coherent and well-motivated framework for iterative drug design. The paper goes beyond simply wrapping an LLM around a search loop: the bi-level Thompson sampling formulation, together with knowledge-guided action selection and refinement, gives the method a more principled structure than many related agentic systems. The reported gains were regarded as substantial, and reviewers also appreciated the additional validation and ablations. The main concerns involved robustness to hallucinations, comparison to non-LLM baselines, and a few exposition issues, but these were largely addressed in rebuttal, and two reviewers explicitly raised their scores after discussion.